# RoCoFT: Efficient Finetuning of Large Language Models with Row-Column Updates

## Abstract

We propose Row-Column Fine-Tuning (RoCoFT), a parameter-efficient fine-tuning method for large-scale language models (LMs) based on updating only a few rows and columns of the weight matrices in transformers. Through extensive experiments with medium size LMs like BERT and RoBERTa, and larger LMs like Bloom-7B, Llama2-7B and Llama2-13B, we show that our method gives comparable or better accuracies than state-of-the-art arameter-Efficient Finetuning methods while also being more memory and computationally-efficient. We also study the reason behind the effectiveness of our method with tools from neural tangent kernel theory. We empirically demonstrate that our kernel, constructed using a restricted set of row and column parameters, is numerically close to the full-parameter kernel and gives comparable classification performance. Ablation studies are conducted to investigate the impact of different algorithmic choices, including the robustness of RoCoFT to any selection of rows and columns, as well as the optimal rank for the effective implementation of our method.

## 1 Introduction

Adapting Large Language Models (LLMs) to different downstream applications is the current prevailing paradigm for solving many Natural Language Processing (NLP) tasks, such as sentiment analysis, machine translation, question answering, named entity recognition, and text summarization. Large language models like GPT-4 (Achiam et al., 2023) and Llama (Touvron et al., 2023) are trained on massive amounts of text data and contain billions of parameters. They give state-of-the-art performance on many NLP, mathematical reasoning (Hendrycks et al., 2021; Cobbe et al., 2021), and programming benchmarks(Jiang et al., 2024). Early works on transfer learning with pretrained LLMs, such as BERT (Devlin et al., 2018) and RoBERTa (Liu et al., 2019), use full fine-tuning, which updates all the parameters of the LLMs when adapting to downstream tasks. This approach becomes impractical as language models continue to scale up (Hoffmann et al., 2022), since a separate copy of the model parameters needs to be stored for each downstream application. Updating all the parameters is also prone to overfitting and the loss of LLM capabilities due to catastrophic forgetting (Kirkpatrick et al., 2017), where the model loses previously learned knowledge while adapting to new tasks.

Adaptor methods (Houlsby et al., 2019) resolve this problem of finetuning LLMs by introducing extra modules called adaptors with a small set of independent parameters. Only the parameters in the adaptors need to be optimized during finetuning, and their small size makes it efficient to adapt an LLM to many different tasks. Parameter-Efficient Finetuning (PEFT) is the study of adapting LLMs to downstream applications by finetuning only a very small set of parameters. Many PEFT methods have been proposed, including the popular LoRA (Hu et al., 2021) and its variants (Zhang et al., 2023b; Edalati et al., 2022; Hyeon-Woo et al., 2021), prefix and prompt tuning (Li & Liang, 2021; Lester et al., 2021), and many other more advanced and complex adaptor methods (He et al., 2021; Zeng et al., 2023). These PEFT methods are effective in reducing the number of parameters required to adapt to downstream tasks, while maintaining performance close to full finetuning.

Despite the numerous PEFT methods available, we pose a critical question: can we design *even simpler* PEFT methods capable of adapting LLMs to diverse downstream tasks in a more efficient way? A simpler method could not only enhance computational and storage efficiency but also offer deeper insights into why PEFT methods succeed as simpler methods are easier to analyze. We answer this

question by presenting a new method called RoCoFT, where the LLMs can be efficiently adapted by updating only a small subset of rows or columns in the transformer block weight matrices. We evaluate the effectiveness of our approach across several benchmarks on different language models. Our experimental results demonstrate that RoCoFT outperforms current PEFT techniques in accuracies, requires fewer trainable parameters and has faster training times. We further analyze our method using Neural Tangent Kernel (NTK) theory (Jacot et al., 2018; Malladi et al., 2023), demonstrating that, for a pretrained LLM, the NTKs derived from a restricted set of rows and columns closely resemble those computed from the full parameter set. This substantiates the effectiveness of our proposed method and further suggests that most of the critical features for fine-tuning are already acquired during the pretraining phase. This insight helps explain why many PEFT methods are effective with only so few parameters, as minimal additional learning is required when a strong set of foundational features is already established. The contributions of this paper are:

• We introduce a new PEFT method called RoCoFT which gives comparable or better accuracies than state-of-the-art PEFT methods, while being more efficient in terms of memory and time complexity. These claims are validated through extensive experiments on language models of different sizes and many benchmark datasets.

• We analyze our method with empirical neural tangent kernels and show that these kernels are close to NTKs defined on the full parameter set, and they give comparable accuracies on many tasks when trained with kernel logistic regression. This explains why our method has performance close to full finetuning from the view of kernel methods.

• We perform extensive ablation studies on the design choices such as which and how many rows and columns to select to facilitate the implementation of our method

## 2 RELATED WORKS

**PEFT Methods:** Parameter-Efficient Finetuning (PEFT) methods aim to finetune only a small number of existing or extra parameters of the LLM to achieve results comparable to finetuning all the parameters. Recently, numerous PEFT approaches have been proposed to advance this strategy. LoRA (Hu et al., 2021) and related methods (Zhang et al., 2023b; Kopiczko et al., 2023; Dettmers et al., 2024) modify existing weight matrices of the model by introducing trainable low-rank decomposition matrices, as adapters, into each layer of the Transformer (Vaswani et al., 2017) architecture. With ranks as low as 4 or 8 for many tasks, they significantly reduce both memory usage and computational time. IA$^3$ (Liu et al., 2022) is another adaptor method that only trains scaling vectors for the key, value, and feed-forward weight matrices in the attention mechanism for task adaptation. Prefix-Tuning (Li & Liang, 2021) and Prompt-Tuning (Lester et al., 2021) work by adding task-specific continuous vectors as contexts for inputs and only updates those parameters while keeping the original LLM parameters frozen. MAM adaptors (He et al., 2021) generalize from both LoRA and prefix-tuning under a unified framework. Our method is closer to LoRA and IA$^3$ in that we modify the weight matrices in the transformer architecture. However, unlike these approaches, we introduce no extra parameters and modify the existing parameters in place. BitFit (Zaken et al., 2021) and LayerNorm Tuning (Zhao et al., 2023) finetune only the bias parameters and layernorm parameters respectively and are extremely parameter-efficient. However, unlike LoRA and our method they cannot increase the capacity of the finetuning model by increasing the rank, since the number of bias and layernorm parameters are fixed in a model.

**Neural Tangent Kernels:** Jacot et al. (2018) and related studies (Lee et al., 2019) show that the training dynamics of an infinite-width multi-layer neural network with suitable Gaussian initialization can be completely described by a fixed kernel called the Neural Tangent Kernel (NTK). This result is further expanded in Yang (2020) to any architecture for which forward and backpropagation can be expressed via nonlinearities and matrix multiplications. In particular, it it shown that in the infinite-width limit for any modern architecture including models with attention layers, the NTK converges almost surely to a deterministic limit. Although these results are asymptotic, this interesting connection between deep neural networks and kernel methods allows many questions about neural networks to be studied via kernels. For example, Wei et al. (2022) studied the generalization error of representations learned by deep neural networks through kernel regression with NTKs. Recently Malladi et al. (2023) proposed to study the effect of finetuning LLMs through their NTKs,

and provided theoretical support to their approach. In this paper, we continue along this line of work to use NTKs to analyze PEFT methods.

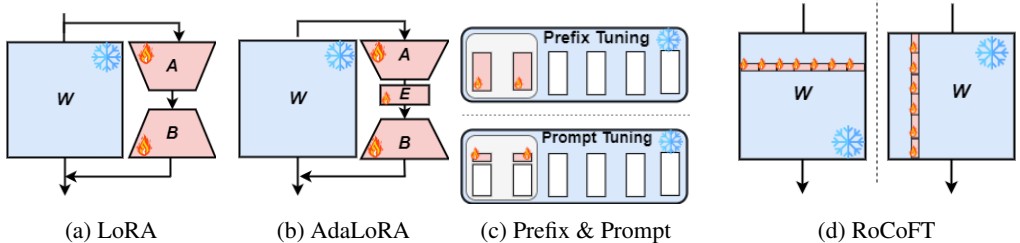

(a) LoRA      (b) AdaLoRA      (c) Prefix & Prompt      (d) RoCoFT

Figure 1: A simplified overview of various PEFT methods and RoCoFT. Snowflake icon indicates frozen parameters while fire icon indicates trainable parameters.

## 3 RoCoFT

PEFT is a collection of methods for transferring pretrained LLMs to downstream tasks by optimizing only a very small set of (additional) parameters. Since most modern LLMs are based on the transformer architecture (Vaswani et al., 2017), there is a line of work in PEFT focusing on modifying the transformer block by freezing most of its parameters and training only a few additional parameters. The method proposed in Houlsby et al. (2019) adds adaptive layers to the transformer blocks, and only parameters in those adaptive layers need to be trained for effective transfer learning. LoRA (Hu et al., 2021) removes the need for adding adaptive layers by directly modifying the weight matrices used in the transformer blocks. There are multiple linear weight matrices in the transformer block taking up most of the parameters, including $\mathbf{W}_q, \mathbf{W}_k, \mathbf{W}_v$ for the query, key and value matrices in the attention mechanism, and also the weights $\mathbf{W}_{ff}$ for the MLP projection layers. LoRA makes use of a low-rank modification of these weight matrices

$$\mathbf{W} = \mathbf{W}_0 + \mathbf{BA}, \tag{1}$$

where $\mathbf{W}_0$ is the pretrained weight matrix, $\mathbf{B}$ and $\mathbf{A}$ are low rank matrices of rank $r$, and $\mathbf{W}$ is the weight matrix after finetuning. In this formulation only $\mathbf{B}$ and $\mathbf{A}$ are updated. If $\mathbf{W}$ is of dimensions $d \times k$, $\mathbf{B}$ and $\mathbf{A}$ will be of dimensions $d \times r$ and $r \times k$ respectively. If $r \ll d, k$, this can lead to significant savings in terms of memory and runtime. IA$^3$ in (Liu et al., 2022) also modifies the weight matrices in the transformer block, but instead of a low-rank modification they rescale the key and value matrices, and also the MLP layers using three learned vectors $\mathbf{l}_k, \mathbf{l}_v$ and $\mathbf{l}_{ff}$ dedicated to key, value and feedforward layers.

The success of these PEFT methods leads us to ask if there are *even simpler* methods for modifying the transformer block for effective finetuning. Here, we propose modifying only a few rows or columns in the weight matrices of the transformer block, for query, key, value weight matrices $\mathbf{W}_q$, $\mathbf{W}_k, \mathbf{W}_v$ and also the weight matrices in the feedforward layer $\mathbf{W}_{ff}$. These can be expressed as

$$\mathbf{W} = \mathbf{W}_0 + \mathbf{R} \qquad \text{and} \qquad \mathbf{W} = \mathbf{W}_0 + \mathbf{C}, \tag{2}$$

where $\mathbf{R}$ and $\mathbf{C}$ are restricted weight matrices such that only at most $r$ of the rows or columns are non-zero. In practice we don't need to form these extra parameters $\mathbf{R}$ and $\mathbf{C}$ and can directly update the parameters in place. Our method is the same as LoRA in its flexibility with increasing the capacity of the finetuning model by increasing the rank $r$, but it is simpler since there is no multiplication of low-rank matrices and all the parameters can be updated in place. There is also no need to worry about the initializations of $\mathbf{A}$ and $\mathbf{B}$, as studied in (Hayou et al., 2024). We call our method RoCoFT for **Ro**w and **Co**lumn-based **F**ine-**T**uning. See Figure 1 for an illustrative diagram.

**Intuition on why the method works:** One might wonder why such a simple update scheme can be effective for finetuning LLMs for different tasks, and our experiments in the following sections show even updating 1 row or 1 column per weight matrix can be extremely effective. We believe this is related to the phenomenon that most of the knowledge of the LLMs are learned during the pretraining stage, and only very limited learning or adaptation is required during finetuning, as observed in the work Zhou et al. (2024) on the limited number of examples required for finetuning/alignment. In

the Appendix we conduct extra experiments to show that even updating a set of randomly selected entries of the weight matrices can work very well. So there is perhaps nothing special about using low-rank or row and column-based matrices. As long as there are sufficient number of parameters spread throughout the LLM for updates, finetuning can be successful. We also give support to this argument of most knowledge already acquired during pretraining by making use of the recent theory of NTK for finetuning LLMs (Malladi et al., 2023; Jacot et al., 2018). We show that the kernels (and thus features) defined by the full parameter set and our restricted parameter set based on a few rows or columns are very similar in their numerical values and classification performance.

## 4 EXPERIMENTS

We evaluate the effectiveness of the proposed RoCoFT method across various NLP tasks, including the General Language Understanding Evaluation (GLUE) benchmark, question answering, text summarization, common sense reasoning, and mathematical reasoning.

**Baselines:** For our baseline comparisons, we utilize prominent PEFT methods such as Adapter (Houlsby et al., 2019), Prompt Tuning (Lester et al., 2021), Prefix-Tuning (Li & Liang, 2021), (IA)$^3$ (Liu et al., 2022), Bitfit (Zaken et al., 2021), LoRA (Hu et al., 2021), AdaLoRA (Zhang et al., 2023a), MAM Adapter (He et al., 2021), PROPETL (Zeng et al., 2023), LoKr (Edalati et al., 2022), (Wu et al., 2024) and LoHa (Hyeon-Woo et al., 2021). The experimental setup for the GLUE benchmark follows Xu et al. (2023), while question answering and text summarization tasks are conducted according to Zhang et al. (2023a).

**Datasets and Model Selection:** For the GLUE benchmark, we evaluate our RoCoFT method on a diverse set of tasks, including CoLA, SST-2, MRPC, STS-B, QQP, MNLI, QNLI, and RTE from Wang et al. (2018), using both RoBERTa Base and Large models (Liu et al., 2019). For question answering, we utilize the SQuAD v1.1 (Rajpurkar et al., 2016) and SQuAD v2.0 (Rajpurkar et al., 2018) datasets with DeBERTa Base v3 (He et al., 2020). Text summarization is evaluated using the XSum (Narayan et al., 2018) and CNN/DailyMail (Hermann et al., 2015) datasets with the BART Large model (Lewis et al., 2019).

For LLM performance using RoCoFT, we conduct an extensive evaluation across thirteen benchmark datasets, covering both common sense reasoning and mathematical reasoning tasks, utilizing four LLMs: Bloom 7B (Le Scao et al., 2023), GPT-J 6B (Wang, 2021), LLaMa2-7B and LLaMA2-13B from Touvron et al. (2023). For common sense reasoning, we employ a wide range of datasets, including BoolQ (Clark et al., 2019), PIQA (Bisk et al., 2020), SIQA (Sap et al., 2019), HellaSwag (Zellers et al., 2019), WinoGrande (Sakaguchi et al., 2021), ARC-easy and ARC-challenge (Clark et al., 2018), and OBQA (Mihaylov et al., 2018), ensuring comprehensive coverage of the model's ability to handle diverse aspects of common sense reasoning. For mathematical reasoning, we use several specialized datasets, including MultiArith (Roy & Roth, 2016), GSM8K (Cobbe et al., 2021), AddSub (Hosseini et al., 2014), SingleEq (Koncel-Kedziorski et al., 2015), and SVAMP (Patel et al., 2021), to assess the model's performance on arithmetic reasoning tasks. Detailed hyperparameter settings are provided in Appendix B. The implementation, environment setup, and hardware details of the experiments are given in Appendix C.

**Performance Analysis:** Table 1 presents the performance of RoCoFT compared with baselines on the GLUE benchmark tasks (Wang et al., 2018). RoCoFT$_{r\text{-Row(Column)}}$ finetunes the model according to equation (2), where in $\mathbf{R}$ and $\mathbf{C}$ the first $r$ rows(columns) are nonzero, respectively. RoCoFT achieves competitive or superior results while updating significantly fewer parameters. For instance, RoCoFT$_{3\text{-Row}}$, with only 0.249 million trainable parameters on RoBERTa-base (Liu et al., 2019) outperforms methods like LoRA (Hu et al., 2021) and MAM Adapter (He et al., 2021), which utilize more parameters. Moreover, RoCoFT variants consistently rank among the top performers across multiple tasks such as the MRPC, QNLI, and RTE, demonstrating robustness and versatility. On RoBERTa-large, RoCoFT$_{3\text{-Row}}$ matches the highest MCC score of 67.39 on the CoLA, achieves an accuracy of 96.69% on the SST-2, and attains better performance on MRPC, Quora Question Pairs (QQP), QNLI, and RTE.

As shown in Table 2, our proposed methods demonstrate superior performance on both question answering and summarization tasks while utilizing significantly fewer trainable parameters. Specifically, on the SQuAD v1.1 dataset (Rajpurkar et al., 2016), the RoCoFT$_{3\text{-Row}}$ method achieves the

| LM | PEFT Method | # TTPs | CoLA | SST2 | MRPC | STS-B | QQP | MNLI | QNLI | RTE |
|---|---|---|---|---|---|---|---|---|---|---|
| RoBERTa$_{\text{Base}}$ | FT | 124.6M | 59.84 | 92.89 | 85.24/88.18 | 90.48/90.16 | 90.18/87.02 | 86.27 | 91.17 | 72.43 |
| | Adapter$^S$ | 7.41M | 61.53 | 94.11 | **89.81/90.85** | 90.25/90.09 | 89.81/**86.90** | 86.27 | 92.06 | 73.56 |
| | Prompt tuning | 0.61M | 49.37 | 92.09 | 70.83/81.72 | 82.44/83.11 | 82.99/78.35 | 80.57 | 80.03 | 58.12 |
| | Prefix-tuning | 0.96M | 59.31 | 93.81 | 84.25/85.03 | 88.48/88.32 | 87.75/84.09 | 85.21 | 90.77 | 54.51 |
| | (IA)$^3$ | 0.66M | 58.58 | 93.92 | 83.00/85.52 | 90.30/90.32 | 87.99/84.10 | 83.95 | 90.88 | 71.12 |
| | BitFit | 0.083M | 61.32 | 93.12 | 87.22/88.41 | 90.34/90.27 | 88.12/84.11 | 84.64 | 91.09 | 77.98 |
| | LoRA | 0.89M | 60.09 | 93.31 | 86.50/88.68 | 90.66/90.47 | 88.83/85.21 | 86.54 | 92.02 | 74.92 |
| | AdaLoRA | 1.03M | 59.82 | 93.92 | 86.49/88.03 | 90.83/90.73 | 88.58/84.98 | 86.26 | 91.43 | 70.04 |
| | MAM Adapter | 1.78M | 58.34 | 94.24 | 87.31/88.21 | 90.74/90.42 | 88.31/83.20 | 86.63 | 90.19 | 72.62 |
| | PROPETL$_{\text{Adapter}}$ | 1.87M | **64.24** | 93.85 | 87.15/87.82 | 90.33/**90.64** | 89.22/85.79 | 86.49 | 91.56 | 75.54 |
| | PROPETL$_{\text{Prefix}}$ | 10.49M | 60.11 | 93.63 | 86.73/87.98 | 90.30/90.19 | 88.54/85.05 | 86.22 | 91.51 | 63.31 |
| | PROPETL$_{\text{LoRA}}$ | 1.77M | 57.94 | 94.11 | 87.42/88.87 | 90.66/90.35 | 88.90/85.55 | 86.83 | 92.04 | 67.39 |
| | MoSLoRA | 1.67M | 60.57 | 93.95 | 86.74/87.98 | 90.05/89.43 | 88.76/85.62 | **87.84** | 90.60 | 75.10 |
| | RoCoFT$_{\text{1-Row}}$ | 0.083M | 60.18 | 94.06 | 87.74/88.48 | 90.70/90.47 | 88.49/85.35 | 85.23 | 90.70 | 76.61 |
| | RoCoFT$_{\text{3-Row}}$ | 0.249M | 63.53 | **94.92** | 89.71/90.74 | **90.89**/90.49 | **89.97**/86.80 | 86.73 | 92.12 | 78.31 |
| | RoCoFT$_{\text{1-Column}}$ | 0.083M | 60.32 | 93.88 | 88.38/89.78 | 90.23/90.14 | 88.46/85.84 | 85.35 | 90.58 | 76.74 |
| | RoCoFT$_{\text{3-Column}}$ | 0.249M | 62.95 | 94.69 | 89.18/90.94 | 90.85/90.45 | 89.86/86.38 | 86.76 | 91.89 | **79.21** |
| RoBERTa$_{\text{large}}$ | FT | 355.3M | 65.78 | 95.50 | 92.22/94.28 | 91.74/91.96 | 90.83/88.68 | 89.21 | 93.19 | 81.40 |
| | Adapter$^S$ | 19.77M | 65.33 | 96.37 | 89.88/90.23 | **92.58**/92.42 | 91.19/87.11 | 91.00 | 94.31 | 85.25 |
| | Prompt-tuning | 1.07M | 61.13 | 94.61 | 73.04/76.29 | 78.51/78.99 | 80.74/75.16 | 68.15 | 89.13 | 60.29 |
| | Prefix-tuning | 2.03M | 59.01 | 95.76 | 88.24/89.37 | 90.92/91.07 | 88.88/85.45 | 89.30 | 93.32 | 74.01 |
| | (IA)$^3$ | 1.22M | 61.15 | 94.61 | 86.45/87.53 | 92.22/86.25 | 89.45/86.25 | 88.63 | 94.25 | 81.23 |
| | Bitfit | 0.222M | 67.01 | 96.10 | 90.93/92.13 | 91.93/**93.38** | 89.48/86.43 | 90.76 | 94.47 | 87.73 |
| | LoRA | 1.84M | 64.47 | **96.67** | 87.50/88.19 | 91.66/91.44 | 90.15/86.91 | 90.76 | 95.00 | 79.78 |
| | AdaLoRA | 2.23M | 65.85 | 94.95 | 89.46/90.34 | 92.05/91.80 | 89.60/86.30 | 90.36 | 94.62 | 77.98 |
| | MAM Adapter | 4.20M | **67.39** | 95.81 | 90.12/92.07 | 92.44/92.18 | 90.87/86.65 | 90.62 | 94.31 | 86.62 |
| | PROPETL$_{\text{Adapter}}$ | 5.40M | 65.55 | 96.27 | 89.71/91.15 | 91.92/91.67 | 90.67/87.74 | **91.37** | 94.80 | 87.69 |
| | PROPETL$_{\text{Prefix}}$ | 26.85M | 62.24 | 96.17 | 90.04/91.92 | 90.70/90.49 | 89.30/86.30 | 90.33 | 94.73 | 79.71 |
| | PROPETL$_{\text{LoRA}}$ | 4.19M | 61.90 | 95.93 | 87.31/89.87 | 91.66/91.38 | 90.93/88.05 | 90.53 | 94.93 | 83.57 |
| | MoSLoRA | 3.23M | 67.27 | 96.17 | 89.96/92.67 | 90.12/87.68 | | 90.29 | 94.73 | 82.41 |
| | RoCoFT$_{\text{1-Row}}$ | 0.222M | 65.70 | 96.63 | 89.97/90.79 | 91.81/92.07 | 90.17/86.15 | 90.73 | 94.20 | 85.31 |
| | RoCoFT$_{\text{3-Row}}$ | 0.666M | **67.39** | **96.69** | **91.05/92.19** | 92.10/92.10 | 90.82/86.11 | 90.98 | 94.85 | 87.83 |
| | RoCoFT$_{\text{1-Column}}$ | 0.222M | 64.89 | 96.60 | 89.12/90.24 | 91.96/92.10 | 90.17/85.83 | 90.81 | 94.17 | 85.71 |
| | RoCoFT$_{\text{3-Column}}$ | 0.666M | 67.18 | 96.67 | 89.88/91.47 | 92.52/92.31 | **91.38/87.12** | 91.13 | 94.85 | 87.82 |

Table 1: RoBERTa models performance on GLUE tasks: Metrics used are MCC for CoLA, accuracy for SST-2, accuracy/F1 score for MRPC and QQP, Pearson/Spearman correlations for STS-B, and accuracy for MNLI, QNLI, and RTE.

| PEFT Method | DeBERTaV3-base | | | BART-large | | |
|---|---|---|---|---|---|---|
| | #TTPs | SQuADv1.1 | SQuADv2.0 | #TTPs | XSum | CNN/DailyMail |
| FT | 184M | 82.83 / 88.14 | 82.92 / 83.75 | 460M | 40.73 / 16.19 / 30.13 | 39.16 / 18.92 / 37.04 |
| Prompt tuning | 0.650M | 74.52 / 78.42 | 73.59 / 76.72 | 0.755M | 38.24 / 14.46 / 27.89 | 37.42 / 17.43 / 34.92 |
| Prefix-tuning | 1.733M | 78.38 / 82.94 | 74.94 / 79.04 | 2.983M | 38.24 / 15.16 / 28.84 | 38.32 / 17.72 / 35.76 |
| LoKr | 0.815M | 80.64 / 86.45 | 80.14 / 81.96 | 1.089M | 39.03 / 16.14 / 30.42 | **40.83** / 19.10 / 38.75 |
| Bitfit | 0.172M | 80.53 / 86.25 | 79.06 / 83.75 | 0.672M | 39.10 / 16.87 / 30.43 | 39.93 / 18.12 / 38.85 |
| LoHa | 0.765M | 81.43 / 88.02 | 81.67 / 85.01 | 1.285M | 40.12 / 18.08 / **32.39** | 39.98 / 18.84 / 38.01 |
| LoRA | 0.740M | 81.64 / 87.16 | 82.56 / 85.75 | 1.242M | **40.63** / 18.44 / 32.15 | 40.74 / 19.10 / 39.24 |
| AdaLoRA | 0.810M | 81.16 / 87.75 | 82.63 / **85.82** | 1.663M | 40.95 / 18.28 / 31.84 | 40.53 / 18.24 / **39.63** |
| RoCoFT$_{\text{Row}}$ | 0.161M | **81.70 / 88.15** | 82.76 / 85.14 | 0.597M | 40.12 / 18.48 / 31.93 | **40.83** / **19.12** / 39.55 |
| RoCoFT$_{\text{Column}}$ | 0.161M | 81.63 / 88.11 | 82.60 / 85.05 | 0.597M | 40.62 / **18.54** / **32.17** | 40.18 / 19.10 / 39.21 |

Table 2: Results of DeBERTaV3-base on SQuAD v1.1, v2.0 benchmarks, reported using EM/F1 scores and BART-large on XSum and CNN/Daily Mail, reported using ROUGE metrics as ROUGE-1/ROUGE-2/ROUGE-L.

highest Exact Match (EM)/F1 scores of 81.70/88.15, outperforming other PEFT methods such as LoRA and AdaLoRA (Zhang et al., 2023b), which require more parameters. Similarly, on SQuAD v2.0 (Rajpurkar et al., 2018), the RoCoFT$_{\text{3-Column}}$ attains the top ROUGE-2 score of 18.54 on XSum, showcasing its effectiveness in handling text summarization.

Table 3 showcases the performance of our proposed RoCoFT across various LLMs and tasks. Notably, these methods consistently achieve superior or competitive results compared to existing PEFT techniques. For the BLOOMZ$_{7B}$ model (Muennighoff et al., 2022), the RoCoFT$_{\text{3-Row}}$ method attains the highest accuracy on Social IQa (SIQA, 73.56%), AI2 Reasoning Challenge (ARC-C, 57.48%), OpenBookQA (OBQA, 72.92%), MultiArith (M.Ar., 79.76%), Arithmetic Sequence (A.S., 70.95%), and Single-Math Problems (S.MP, 54.42%). The RoCoFT$_{\text{3-Column}}$ variant also performs exceptionally well, achieving top scores on WinoGrande (W.Gra., 72.50%) and Grade School Math 8K (GSM8K, 71.05%). Similarly, with the GPT-J$_{6B}$ model (Wang, 2021), our methods maintain strong performance. The RoCoFT$_{\text{3-Row}}$ method achieves the best results on Boolean Questions

| LLM | Method | # TTPs | BoolQ | PIQA | SIQA | H.Sw. | W.Gra. | ARCe | ARCc | OBQA | M.Ar. | G.8K | A.S. | S.eEq | S.MP |
|---|---|---|---|---|---|---|---|---|---|---|---|---|---|---|---|
| BLOOMZ$_{7B}$ | Prefix | 33.37M | 58.53 | 62.24 | 65.41 | 48.32 | 66.63 | 68.13 | 49.32 | 63.51 | 78.41 | 66.45 | 67.52 | 66.94 | 49.10 |
| | AdaLoRA | 24.88M | 64.94 | **74.68** | 72.49 | 55.89 | 68.30 | 73.21 | 56.59 | 72.85 | 79.43 | 70.25 | 68.93 | **70.93** | 53.89 |
| | (IA)³ | 19.34M | 63.30 | _73.33_ | 71.01 | 52.50 | 71.60 | 69.45 | 54.14 | 68.60 | 78.90 | **71.17** | 70.33 | 70.84 | 53.95 |
| | LoRA | 24.22M | 65.89 | 73.92 | 73.33 | **56.65** | 71.39 | **73.46** | 57.15 | 72.31 | 79.50 | 70.93 | _70.90_ | 70.59 | 53.85 |
| | RoCoFT₃-Row | 13.37M | _66.33_ | 74.53 | **73.56** | _56.60_ | 72.14 | _73.29_ | **57.48** | 72.92 | **79.76** | 70.94 | **70.95** | _70.90_ | **54.42** |
| | RoCoFT₃-Column | 13.37M | **66.34** | _74.64_ | 73.12 | 55.93 | **72.50** | 73.11 | _57.19_ | _72.90_ | _79.72_ | 71.05 | 70.88 | 70.76 | _54.38_ |
| GPT-J$_{6B}$ | Prefix | 27.83M | 62.28 | 65.04 | 67.72 | 44.15 | 63.71 | 63.59 | 46.47 | 58.31 | 83.12 | 67.44 | 75.25 | 78.46 | 49.12 |
| | AdaLoRA | 20.77M | 65.19 | 67.58 | **71.22** | 45.16 | 66.03 | 64.10 | **47.75** | 63.92 | 88.51 | 72.45 | 80.21 | _82.03_ | 56.14 |
| | (IA)³ | 16.61M | 63.17 | _68.51_ | 68.97 | 45.79 | 66.06 | 62.42 | 45.32 | **65.42** | _89.51_ | 72.04 | _80.50_ | 81.50 | 55.43 |
| | LoRA | 20.02M | _65.50_ | 67.63 | 69.46 | 45.60 | _66.80_ | 63.56 | 46.81 | 63.82 | 88.30 | **72.82** | **80.60** | 81.24 | _56.73_ |
| | RoCoFT₃-Row | 11.62M | **65.92** | 68.53 | 69.90 | _45.97_ | **66.87** | **64.91** | 45.12 | _65.07_ | 89.45 | _72.80_ | 80.45 | 82.12 | **56.79** |
| | RoCoFT₃-Column | 11.62M | 65.12 | 68.22 | _69.96_ | **45.98** | 66.78 | _64.89_ | 45.70 | 64.81 | **89.74** | 72.24 | 80.23 | **82.61** | 56.70 |
| LLaMA-2$_{7B}$ | Prefix | 33.53M | 67.33 | 79.46 | 75.80 | 76.04 | 72.11 | 71.67 | 57.33 | 69.98 | 84.18 | 68.47 | 81.04 | 80.00 | 52.17 |
| | AdaLoRA | 24.90M | 67.03 | 78.69 | 76.06 | 88.85 | 76.47 | _76.50_ | 60.36 | 74.22 | 89.81 | 77.07 | **86.70** | _83.01_ | 60.25 |
| | (IA)³ | 19.42M | 65.02 | 78.10 | _78.00_ | 87.57 | 76.78 | _76.78_ | 60.54 | 74.02 | 90.20 | 76.13 | _86.55_ | **83.70** | 59.16 |
| | LoRA | 24.30M | 67.09 | 79.37 | 76.15 | 88.86 | **77.54** | **76.54** | _60.55_ | 74.63 | 90.13 | 75.68 | 84.67 | 82.14 | 59.94 |
| | RoCoFT₃-Row | 13.47M | **69.36** | _80.01_ | **78.09** | _89.28_ | 76.73 | 76.46 | _60.55_ | **76.96** | **90.55** | **77.37** | 86.12 | 82.66 | **60.75** |
| | RoCoFT₃-Column | 13.47M | _69.32_ | **80.08** | 77.99 | **89.46** | 76.41 | 76.46 | **60.59** | _76.90_ | _90.42_ | _77.35_ | 86.16 | 82.48 | 60.35 |
| LLaMA-2$_{13B}$ | Prefix | 61.97M | 68.38 | 80.99 | 77.80 | 80.00 | 76.35 | 77.62 | 61.32 | 72.94 | 87.22 | 71.09 | 84.09 | 81.28 | 58.25 |
| | AdaLoRA | 45.04M | **71.71** | 82.55 | 78.88 | 91.60 | 83.01 | 83.04 | **67.33** | **81.76** | 90.55 | **80.19** | 87.00 | 87.10 | 66.03 |
| | (IA)³ | 36.02M | 71.39 | 83.33 | 78.32 | **92.40** | 83.24 | 83.34 | 66.43 | 80.99 | **91.88** | 79.24 | _88.16_ | 87.08 | 65.63 |
| | LoRA | 44.94M | 71.19 | **83.99** | 79.15 | _91.86_ | **83.24** | 83.35 | 67.05 | 81.37 | 91.27 | 78.90 | 86.89 | 86.07 | 65.85 |
| | RoCoFT₃-Row | 24.88M | _71.46_ | 83.32 | **79.54** | _91.86_ | 83.22 | **83.65** | _67.12_ | 81.54 | 90.69 | _79.70_ | **88.24** | _87.28_ | **66.60** |
| | RoCoFT₃-Column | 24.88M | 71.44 | _83.52_ | _79.50_ | 91.84 | _83.20_ | _83.39_ | 67.06 | _81.73_ | _91.46_ | 79.63 | 88.11 | **87.58** | _66.63_ |

Table 3: Accuracy comparison of commonsense and mathematical reasoning performance across different PEFT methods using LLMs.

(BoolQ, 65.92%), MultiArith (89.45%), and S.MP (56.79%), while the RoCoFT₃-Column method excels on SIQA (69.96%) and SingleEq (S.eEq, 82.61%).

When scaled to larger models like LLaMA2$_{7B}$ and LLaMA2$_{13B}$ (Touvron et al., 2023), our methods continue to demonstrate their effectiveness. On LLaMA2$_{7B}$, the RoCoFT₃-Row method secures the highest accuracy on BoolQ (69.36%), SIQA (78.09%), OBQA (76.96%), M.Ar. (90.55%), and GSM8K (77.37%). The RoCoFT₃-Column variant achieves top performance on HellaSwag (H.Sw., 89.46%) and S.eEq (82.48%). For LLaMA2$_{13B}$, both RoCoFT₃-Row and RoCoFT₃-Column methods attain leading results on multiple tasks, with the RoCoFT₃-Row method achieving the highest accuracy on SIQA (79.54%), ARC-Easy (ARCe, 83.65%), A.S. (88.24%), and S.MP (66.60%).

These results underscore RoCoFT's ability to deliver state-of-the-art performance while maintaining parameter efficiency, making it highly suitable for deployment in resource-constrained environments.

| Methods | Space | Time | TTPs | APs |
|---|---|---|---|---|
| FT | $O(d \times d)$ | $O(d \times d)$ | $d^2$ | 0 |
| (IA)³ | $O(l_k + l_v + l_{ff})$ | $O(d_k + d_v + d_{ff})$ | $3d$ | $3d$ |
| Prompt | $O(d \times l_p)$ | $O(d \times l_p)$ | $l_p.d$ | $l_p.d$ |
| Prefix | $O(L \times d \times l_p)$ | $O(L \times d \times l_p)$ | $L.l_p.d$ | $L.l_p.d$ |
| LoRA | $O((d+d) \times r)$ | $O((d+d) \times r)$ | $2dr$ | $2dr$ |
| LoRA-FA | $O((d+d) \times r)$ | $O((d+d) \times r)$ | $dr$ | $2dr$ |
| AdaLoRA | $O((d+d+r) \times r)$ | $O((d+d+r) \times r)$ | $2dr+r^2$ | $2dr+r^2$ |
| LoHA | $O(2r \times (d+d))$ | $O(2r \times (d+d))$ | $4dr$ | $4dr$ |
| BitFit | $O(d)$ | $O(d)$ | $d$ | 0 |
| _RoCoFT$_{Row}$_ | $O(d \times r)$ | $O(d \times r)$ | $rd$ | 0 |
| _RoCoFT$_{Column}$_ | $O(d \times r)$ | $O(d \times r)$ | $rd$ | 0 |

Table 4: Space/Time Complexity; Total Trainable Parameters (TTPs) and Additional Parameters in model (Aps) for *RoCoFT* method and baseline methods for single layer $\mathbf{W} \in \mathbb{R}^{d \times d}$. Within this table, we define $l_k, l_v$, and $l_{ff}$ as the dimensions of three learned vectors in IA³; and $l_p$ as the length of the prompt added to the input/layers in prompt tuning and prefix-tuning. For LoRA-type methods, we use $r$ to represent the rank dimension.

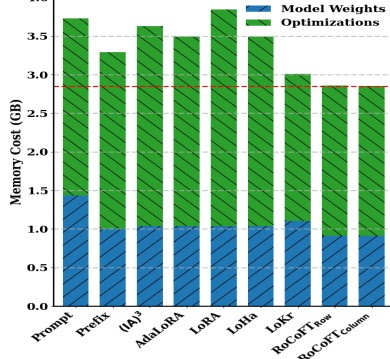

Figure 2: Comparison of memory costs for PEFT Methods. Blue bars show the memory cost of the original model weights, while green bars show the memory cost for optimization in each method.

**Efficiency Comparison:** Our proposed method, RoCoFT, demonstrates significant parameter efficiency compared to existing PEFT techniques. Specifically, RoCoFT variants require substantially fewer trainable parameters while achieving competitive or superior performance.

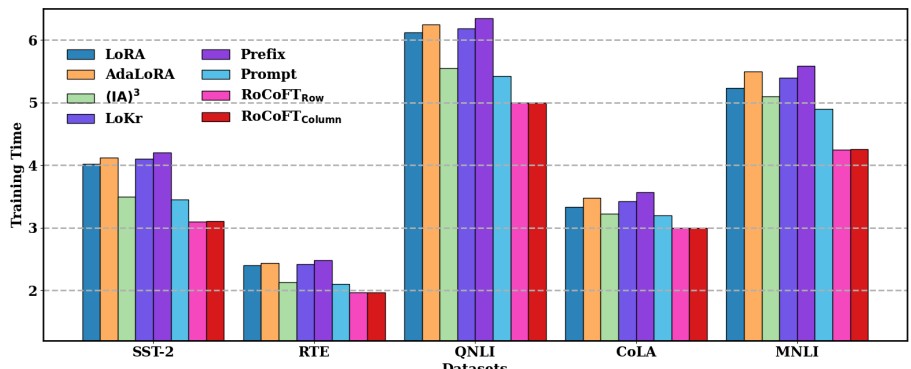

Figure 3: Training time (minutes) comparison across different PEFT methods.

For instance, as shown in Table 1, RoCoFT$_{\text{Row}}$ uses only 0.083 million trainable parameters for rank one and 0.249 million for rank three on the GLUE benchmark (Wang et al., 2018), outperforming methods like LoRA (Hu et al., 2021) and MAM Adapter (He et al., 2021), which use 0.89 million and 1.78 million parameters, respectively. Similarly, in question answering and summarization tasks (Table 2), our Row and Column methods utilize just 0.161 million trainable parameters, significantly less than LoRA and AdaLoRA (Zhang et al., 2023b), yet achieve higher or comparable performance.

In terms of computational efficiency (Table 4), our method exhibits lower space and time complexity. Specifically, RoCoFT has a time/space complexity of $O(r \times d)$, compared to LoRA's $O(2d \times r)$ and Prefix-Tuning's $O(L \times d \times l_p)$, where $r$ is the rank, $d$ is the model dimension, $L$ is the number of layers, and $l_p$ is the length of the prefix. Moreover, our method does not introduce any additional parameters into the model architecture, which also reduces the total number of parameters and requires less GPU memory and training time, as illustrated in Figure 2. RoCoFT variants have lower memory occupancy during training (approximately 2.85GB) compared to other methods like LoRA and AdaLoRA, and consistently require less training time across various datasets, as shown in Figure 3.

These results underscore the efficiency of our approach in terms of both parameter count and computational resources, highlighting its suitability for deployment in resource-constrained environments.

## 5 Finetuning through the lens of Neural Tangent Kernel Regression

Kernel methods are classic machine learning algorithms that make use of kernel functions for learning nonlinear mapping of inputs, with SVMs (Cortes & Vapnik, 1995) and Gaussian Processes (Williams & Rasmussen, 2006) being the prime examples. A kernel function $\mathbf{K} : \mathcal{X} \times \mathcal{X} \to \mathbb{R}$ is a similarity function on the input space $\mathcal{X}$ that satisfies certain symmetry and positive semi-definiteness conditions. If these conditions (Mercer's conditions) are satisfied the kernel behaves like an inner product over a possibly infinite dimensional space called the Reproducing Kernel Hilbert Space (RKHS) $\mathcal{H}$. The RKHS $\mathcal{H}$ contains functions $f : \mathcal{X} \to \mathbb{R}$ that maps inputs to real numbers. Although the functions $f \in \mathcal{H}$ are infinite dimensional, when minimizing a loss function $\mathcal{L}$ over a finite training sample $\{\mathbf{x}_i, \mathbf{y}_i\}_{i=1}^{n}$, the Representer Theorem (Schölkopf et al., 2001) tells us that the optimal solution takes the form

$$f^*(\cdot) = \sum_{i=1}^{n} \alpha_i \mathbf{K}(\mathbf{x}_i, \cdot),$$

where $\alpha_i$ are coefficients that depend on labels $\mathbf{y}_i$. When the loss function $\mathcal{L}$ is convex the minimization problem is also convex. The ability to learn nonlinear mappings of inputs through solving convex optimization problems makes kernel methods a powerful tool in machine learning.

Kernel methods differ from deep learning with neural networks in that the kernels (and hence the feature representations) are fixed during learning, while deep neural networks continuously update their feature representations during backpropagation. Jacot et al. (2018) made the important discovery that under certain conditions, in the infinite width limit, the training of deep neural networks

can be described by a fixed kernel called the Neural Tangent Kernel (NTK). For a neural network function $f_{\boldsymbol{\theta}} : \mathcal{X} \to \mathbb{R}^k$ parameterized by $\boldsymbol{\theta}$, its Neural Tangent Kernel is defined by

$$\mathbf{K}_{\boldsymbol{\theta}}(\mathbf{x}, \mathbf{x}') = \langle \nabla f_{\boldsymbol{\theta}}(\mathbf{x}), \nabla f_{\boldsymbol{\theta}}(\mathbf{x}') \rangle,$$

where $\nabla f_{\boldsymbol{\theta}}(\mathbf{x})$ is the corresponding Jacobian.($\nabla f_{\boldsymbol{\theta}}(\mathbf{x})$ is $p \times k$ if $\boldsymbol{\theta}$ has $p$ parameters, and $\mathbf{K}_{\boldsymbol{\theta}}(\mathbf{x}, \mathbf{x}')$ is $k \times k$) This connection allows us to study the behaviour of neural networks with kernel methods. Malladi et al. (2023) extends the NTK theory to model the finetuning of LLMs. As an alternative to finetuning by SGD, given training data $(\mathbf{x}_i, \mathbf{y}_i)_{i=1}^n$ for a downstream classification task, we can instead solve the following kernel logistic regression problem

$$\min_{f \in \mathcal{H}} \sum_{i=1}^n \mathcal{L}(f(\mathbf{x}_i), \mathbf{y}_i) + \frac{\lambda}{2} \|f\|_{\mathcal{H}}^2,$$

where $\mathcal{H}$ is the RKHS defined by the NTK $\mathbf{K}_{\boldsymbol{\theta}}$, and $\mathcal{L}(\cdot, \cdot)$ is the logistic loss. For a two-class problem with $\mathbf{y}_i \in \{0, 1\}$, this is equivalent to

$$\min_{\boldsymbol{\alpha}} - \sum_{i=1}^n \sum_{j=1}^n \mathbf{y}_i \alpha_j \mathbf{K}_{\boldsymbol{\theta}}(\mathbf{x}_i, \mathbf{x}_j) + \sum_{i=1}^n \log(1 + \exp(\sum_{j=1}^n \alpha_j \mathbf{K}_{\boldsymbol{\theta}}(\mathbf{x}_i, \mathbf{x}_j))) + \frac{\lambda}{2} \sum_{i=1}^n \sum_{j=1}^n \alpha_i \alpha_j \mathbf{K}_{\boldsymbol{\theta}}(\mathbf{x}_i, \mathbf{x}_j).$$

This problem is convex in $\boldsymbol{\alpha}$ so the solution is easy to describe without local minima. It is also clear that the solution is completely determined by the value of the NTK $\mathbf{K}_{\boldsymbol{\theta}}$ between all training samples $\mathbf{x}_i$. Notice that $\boldsymbol{\theta}$ is fixed here (usually set to pretrained model weights), so the kernel $\mathbf{K}_{\boldsymbol{\theta}}$ is also fixed. Malladi et al. (2023) provides theoretical analysis on conditions when finetuning with SGD will converge to this particular kernel logistic regression setup at the infinite width limit.

Under this framework it becomes feasible to compare full finetuning with finetuning over a subset of parameters by comparing their respective NTKs. Below, we compare the kernels of the 1-row and 1-column version of our RoCoFT method, and we denote the associated trainable parameters as $\boldsymbol{\theta}_R, \boldsymbol{\theta}_C \subseteq \boldsymbol{\theta}$. The corresponding kernels are defined as

$$\mathbf{K}_{\boldsymbol{\theta}_R}(\mathbf{x}, \mathbf{x}') = \langle \nabla f_{\boldsymbol{\theta}_R}(\mathbf{x}), \nabla f_{\boldsymbol{\theta}_R}(\mathbf{x}') \rangle \quad \text{and} \quad \mathbf{K}_{\boldsymbol{\theta}_C}(\mathbf{x}, \mathbf{x}') = \langle \nabla f_{\boldsymbol{\theta}_C}(\mathbf{x}), \nabla f_{\boldsymbol{\theta}_C}(\mathbf{x}') \rangle$$

Note that while the Jacobians $\nabla f_{\boldsymbol{\theta}_R}(\mathbf{x})$ and $\nabla f_{\boldsymbol{\theta}_C}(\mathbf{x})$ can have different dimensions due to different number of parameters, the kernels $\mathbf{K}_{\boldsymbol{\theta}_R}$ and $\mathbf{K}_{\boldsymbol{\theta}_C}$ reside in the same function space $\mathcal{H}$ (so does the full finetuning kernel $\mathbf{K}_{\boldsymbol{\theta}}$) and can be compared on a data sample.

We first compare few-shot learning performance of these kernels using kernel logistic regression with prompt-based finetuning, as done in Malladi et al. (2023). The kernels are computed with the pretrained RoBERTa-base model. From Table 5 we can see the performance of kernel logistic regression using $\mathbf{K}_{\boldsymbol{\theta}_R}$ and $\mathbf{K}_{\boldsymbol{\theta}_C}$ are surprisingly close to using the kernel for full parameters $\mathbf{K}_{\boldsymbol{\theta}}$, usually within the standard error of 5 runs using different random seeds. The performance of kernel logistic regression using $\mathbf{K}_{\boldsymbol{\theta}}$ is in turn close to full finetuning except for a few tasks including TREC, MNLI, SNLI, QNLI and MPQA, which are related to the prompt templates used. Next we directly compare the kernel matrices $\mathbf{K}_{\boldsymbol{\theta}}$, $\mathbf{K}_{\boldsymbol{\theta}_R}$ and $\mathbf{K}_{\boldsymbol{\theta}_C}$ for these few-shot learning problems directly. Figure 4 shows the empirical Neural Tangent Kernel values for the the task SST-2. More figures for the other tasks are available in Appendix E. This task is a two-class problem and hence their kernel matrices have 2x2 block structure. The values of the kernel entries are capped at 95-percentile for better visualization under heatmap. We can see that except for the magnitude of the entries in the kernel matrices, the patterns in the kernel matrices for the full parameter set $\mathbf{K}_{\boldsymbol{\theta}}$, 1-row set $\mathbf{K}_{\boldsymbol{\theta}_R}$ and 1-column set $\mathbf{K}_{\boldsymbol{\theta}_C}$ are extremely similar. This is while the NTK for LoRA with $r = 1$ is not as close as the NTK for row/column parameters to the full parameter kernel. More quantitatively, Table 6 shows the relative difference between the 1-row kernel $\mathbf{K}_{\boldsymbol{\theta}_R}$ and 1-column kernel $\mathbf{K}_{\boldsymbol{\theta}_C}$ with the full parameter kernel $\mathbf{K}_{\boldsymbol{\theta}}$ after normalization in $\ell_1$ and $\ell_2$ norms by flattening the kernel matrices. For example, the relative difference for $\mathbf{K}_{\boldsymbol{\theta}_R}$ is computed as

$$\|(\mathbf{K}_{\boldsymbol{\theta}_R}/\|\mathbf{K}_{\boldsymbol{\theta}_R}\|_p) - (\mathbf{K}_{\boldsymbol{\theta}}/\|\mathbf{K}_{\boldsymbol{\theta}}\|_p)\|_p, \quad p = 1, 2.$$

We can see that except for few tasks like MNLI, SNLI and TREC, the relative differences between kernels are between 5 to 15%, which are fairly small. These results across many tasks from NTK provide strong support for our proposal that finetuning only a few rows or columns can give performance comparable to full finetuning.

| k-shot (single) | Method | SST-2 | SST-5 | MR | CR | MPQA | Subj | TREC |
|---|---|---|---|---|---|---|---|---|
| **16** | Full FT | 89.0(1.5) | 44.6(1.4) | 83.2(2.4) | 93.3(0.2) | 83.3(1.3) | 88.5(2.6) | 80.3(7.2) |
| | $\mathbf{K}_{\boldsymbol{\theta}}$ | 88.3(0.3) | 43.6(2.2) | 84.7(1.5) | 93.2(0.9) | 76.4(2.7) | 88.6(1.3) | 56.0(9.2) |
| | $\mathbf{K}_{\boldsymbol{\theta}_R}$ | 88.5(0.4) | 42.9(1.9) | 83.9(1.2) | 93.2(0.5) | 77.3(2.1) | 85.8(1.2) | 51.6(3.9) |
| | $\mathbf{K}_{\boldsymbol{\theta}_C}$ | 88.6(2.4) | 42.4(1.9) | 84.6(1.0) | 93.2(0.5) | 77.6(2.0) | 85.9(1.2) | 51.2(6.7) |
| | $\mathbf{K}_{\text{LoRA}}$ | 88.5(0.7) | | 84.5(1.4) | 93.2(0.5) | | | |
| **64** | Full FT | 89.7(0.4) | 45.8(2.1) | 85.6(1.1) | 94.3(0.5) | 84.8(0.8) | 92.9(0.5) | 93.2(1.0) |
| | $\mathbf{K}_{\boldsymbol{\theta}}$ | 89.2(1.0) | 46.0(1.3) | 86.4(0.6) | 93.7(0.4) | 81.2(0.9) | 91.4(0.7) | 77.8(2.3) |
| | $\mathbf{K}_{\boldsymbol{\theta}_R}$ | 89.5(0.5) | 46.0(1.5) | 86.4(0.6) | 93.9(0.6) | 81.6(0.7) | 90.4(0.4) | 70.4(1.6) |
| | $\mathbf{K}_{\boldsymbol{\theta}_C}$ | 89.5(0.6) | 45.9(1.5) | 86.4(0.4) | 93.9(0.6) | 81.5(0.5) | 90.5(0.6) | 70.7(2.5) |

| k-shot (pair) | Method | MNLI | SNLI | QNLI | RTE | MRPC | QQP |
|---|---|---|---|---|---|---|---|
| **16** | Full FT | 59.2(2.7) | 65.7(2.7) | 62.1(3.1) | 60.0(5.5) | 73.9(2.7) | 62.1(2.3) |
| | $\mathbf{K}_{\boldsymbol{\theta}}$ | 53.0(3.0) | 57.8(2.3) | 60.1(3.3) | 60.0(4.7) | 73.4(5.6) | 58.2(0.9) |
| | $\mathbf{K}_{\boldsymbol{\theta}_R}$ | 51.1(2.8) | 56.0(1.8) | 59.6(2.3) | 58.6(6.0) | 69.3(5.9) | 57.1(3.3) |
| | $\mathbf{K}_{\boldsymbol{\theta}_C}$ | 51.9(2.7) | 56.4(1.8) | 59.2(2.6) | 58.1(5.6) | 69.2(4.7) | 58.4(1.7) |
| | $\mathbf{K}_{\text{LoRA}}$ | | | 59.9(3.0) | 58.8(4.7) | | 58.2(2.6) |
| **64** | Full FT | 68.7(1.7) | 77.3(0.9) | 72.8(2.2) | 68.9(2.5) | 82.8(1.2) | 69.2(1.3) |
| | $\mathbf{K}_{\boldsymbol{\theta}}$ | 60.4(1.8) | 65.5(1.6) | 67.3(1.6) | 66.5(2.5) | 79.2(2.5) | 66.4(1.7) |
| | $\mathbf{K}_{\boldsymbol{\theta}_R}$ | 58.0(2.0) | 64.7(1.0) | 66.2(1.7) | 61.1(0.8) | 72.2(4.5) | 64.2(3.0) |
| | $\mathbf{K}_{\boldsymbol{\theta}_C}$ | 58.4(2.5) | 64.4(1.4) | 66.7(1.8) | 62.7(0.9) | 73.5(4.6) | 64.6(2.4) |

Table 5: Single-sentence and sentence-pair tasks comparing kernels for RoCoFT (1 row and 1 column), kernels for all parameters, and full finetuning.

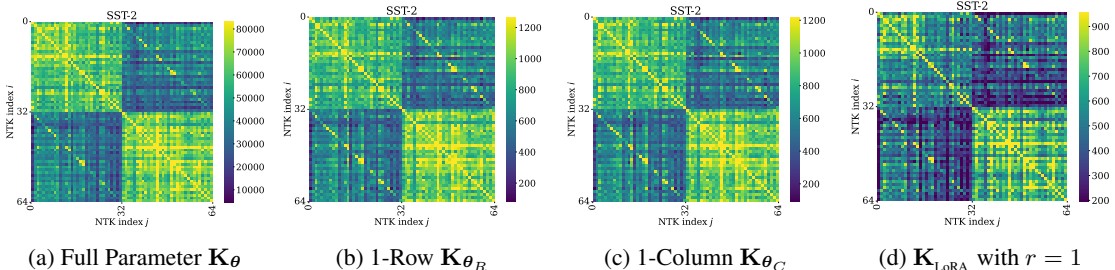

(a) Full Parameter $\mathbf{K}_{\boldsymbol{\theta}}$     (b) 1-Row $\mathbf{K}_{\boldsymbol{\theta}_R}$     (c) 1-Column $\mathbf{K}_{\boldsymbol{\theta}_C}$     (d) $\mathbf{K}_{\text{LoRA}}$ with $r=1$

Figure 4: Neural Tangent Kernels for SST-2 on 16-shot training data.

# 6 ABLATION STUDIES

**Robustness of Row-Column Selection:** In this study, we demonstrate the robustness of our row and column selection method through a detailed comparison of four selection strategies: Max, Min, Mixed, and random. These strategies are applied to both rows and columns of the weight matrices. For the Min, Max, and Mixed selection strategies, we employ a scoring criterion used in the Wanda method (Sun et al., 2023), a simple yet effective pruning technique that requires only the forward pass. Pruning a neural network involves scoring the weights by importance (e.g., by the absolute values of weights), and then remove the least important ones. We can adopt these strategies to rank rows and columns by importance and evaluate the effect of finetuning on them. Given a weight matrix $\mathbf{W} \in \mathbb{R}^{d_{out} \times d_{in}}$ and input feature activations $\mathbf{X} \in \mathbb{R}^{s \times d_{in}}$ from a length $s$ sequence, Wanda calculates the importance score $\mathbf{S}_{ij}$ of the weight $\mathbf{W}_{ij}$ as

$$\mathbf{S}_{ij} = |\mathbf{W}_{ij}| \cdot \|\mathbf{X}_{\cdot j}\|_2, \tag{3}$$

| 16-shot (single) | SST-2 | SST-5 | MR | CR | MPQA | Subj | TREC |
|---|---|---|---|---|---|---|---|
| $\mathbf{K}_{\boldsymbol{\theta}_R}, p\!=\!1$ | 0.093(0.008) | 0.083(0.005) | 0.064(0.006) | 0.087(0.007) | 0.123(0.012) | 0.061(0.005) | 0.181(0.007) |
| $\mathbf{K}_{\boldsymbol{\theta}_R}, p\!=\!2$ | 0.130(0.014) | 0.113(0.012) | 0.092(0.011) | 0.126(0.021) | 0.182(0.017) | 0.073(0.008) | 0.197(0.008) |
| $\mathbf{K}_{\boldsymbol{\theta}_C}, p\!=\!1$ | 0.091(0.008) | 0.077(0.004) | 0.061(0.006) | 0.084(0.006) | 0.123(0.014) | 0.055(0.005) | 0.166(0.007) |
| $\mathbf{K}_{\boldsymbol{\theta}_C}, p\!=\!2$ | 0.127(0.016) | 0.108(0.012) | 0.089(0.011) | 0.122(0.018) | 0.184(0.018) | 0.069(0.009) | 0.185(0.008) |
| **16-shot (pair)** | **MNLI** | **SNLI** | **QNLI** | **RTE** | **MRPC** | **QQP** | |
| $\mathbf{K}_{\boldsymbol{\theta}_R}, p\!=\!1$ | 0.177(0.011) | 0.198(0.039) | 0.076(0.028) | 0.140(0.019) | 0.073(0.009) | 0.046(0.008) | |
| $\mathbf{K}_{\boldsymbol{\theta}_R}, p\!=\!2$ | 0.260(0.043) | 0.255(0.069) | 0.149(0.071) | 0.203(0.039) | 0.096(0.016) | 0.063(0.013) | |
| $\mathbf{K}_{\boldsymbol{\theta}_C}, p\!=\!1$ | 0.176(0.013) | 0.194(0.040) | 0.073(0.028) | 0.142(0.023) | 0.073(0.010) | 0.044(0.006) | |
| $\mathbf{K}_{\boldsymbol{\theta}_C}, p\!=\!2$ | 0.262(0.050) | 0.253(0.072) | 0.146(0.071) | 0.212(0.047) | 0.096(0.016) | 0.061(0.011) | |

Table 6: Relative difference in kernels (compared to full parameter $\mathbf{K}_{\boldsymbol{\theta}}$) on single-sentence and sentence-pair tasks.

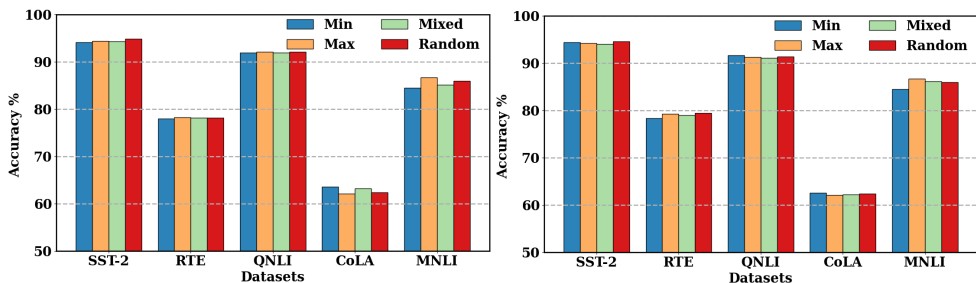

Figure 5: Accuracy comparison of Max, Min, Mixed, and random row and column selection methods across different datasets. The results show that the proposed selection techniques are robust across various tasks.

where $\|\mathbf{X}_{\cdot j}\|_2$ is the 2-norm across the $j$th feature aggregated across all examples in batch. To determine the most important rows, we sum $\mathbf{S}_{ij}$ across the columns, yielding a row score vector $\mathbf{S}_{\text{row}} \in \mathbb{R}^{d_{in}}$. The rows are then sorted by this score, and we select the top $r$ rows according to either the Max or Min scores. The same procedure is applied to columns by summing across the rows, producing a column score $\mathbf{S}_{\text{column}} \in \mathbb{R}^{d_{out}}$. The Mixed strategy takes half of the rows/columns from Min and half from Max, while the random strategy selects rows and columns uniformly at random.

Figure 5 presents the comparative results of these four strategies on the SST-2, RTE, QNLI, CoLA, and MNLI datasets for rank $r = 4$. Across all datasets, the results show consistent robustness, indicating that our method performs well regardless of the selection criteria—whether based on Max, Min, MinMax, or random selection of rows or columns.

**Optimal Rank $r$ for RoCoFT :** We investigate the impact of varying the rank $r$ on the performance of RoCoFT (Row and Column Fine-Tuning) and compare it with the widely used LoRA method within the RoBERTa-base attention block. We assess key metrics such as training time, accuracy, number of parameters, and memory consumption for each rank $r \in \{1, 2, 4, 8, 64\}$ using the SST2 dataset. The results are summarized in Table 7.

From the table, we observe that as the rank $r$ increases, both RoCoFT and LoRA exhibit improved accuracy. For lower ranks, such as $r = 1$ and $r = 2$, RoCoFT$_{\text{row}}$ and RoCoFT$_{\text{column}}$ consistently outperform LoRA in terms of both training time and parameter efficiency, while maintaining competitive accuracy. Specifically,

| Rank | Algorithm | Time | Accuracy | Parameters | Memory |
|---|---|---|---|---|---|
| | LoRA | 3:12 | 0.910 | 0.055 | 2762 |
| 1 | RoCoFT$_{\text{row}}$ | 3:00 | 0.913 | 0.022 | 2372 |
| | RoCoFT$_{\text{column}}$ | 2:59 | 0.912 | 0.022 | 2373 |
| | LoRA | 3:25 | 0.922 | 0.110 | 2768 |
| 2 | RoCoFT$_{\text{row}}$ | 3:00 | 0.920 | 0.055 | 2410 |
| | RoCoFT$_{\text{column}}$ | 3:00 | 0.922 | 0.055 | 2414 |
| | LoRA | 3:27 | 0.925 | 0.221 | 2771 |
| 4 | RoCoFT$_{\text{row}}$ | 3:01 | 0.923 | 0.110 | 2450 |
| | RoCoFT$_{\text{column}}$ | 3:01 | 0.922 | 0.110 | 2451 |
| | LoRA | 3:29 | 0.929 | 0.442 | 2783 |
| 8 | RoCoFT$_{\text{row}}$ | 3:03 | 0.930 | 0.221 | 2336 |
| | RoCoFT$_{\text{column}}$ | 3:02 | 0.928 | 0.221 | 2335 |
| | LoRA | 3:33 | 0.928 | 3.538 | 2993 |
| 64 | RoCoFT$_{\text{row}}$ | 3:06 | 0.934 | 1.769 | 2656 |
| | RoCoFT$_{\text{column}}$ | 3:05 | 0.933 | 1.769 | 2653 |

Table 7: Comparison with LoRA in terms of rank, training time (minutes), accuracy, number of parameters, and memory usage (MB).

for rank $r = 1$, RoCoFT$_{\text{row}}$ achieves an accuracy of 0.913 while using only 0.022 million parameters, which is significantly fewer than LoRA's 0.055 million parameters for the same rank, with a slight increase in accuracy. This demonstrates the parameter efficiency of RoCoFT at lower ranks.

As the rank increases to $r = 8$, both RoCoFT variants continue to show slight improvements in accuracy while maintaining a faster training time compared to LoRA. Notably, at higher ranks like $r = 64$, RoCoFT$_{\text{row}}$ achieves the highest accuracy of 0.934 with a significantly lower memory footprint compared to LoRA (2.656 GB vs. 2.993 GB).

# 7 CONCLUSIONS

We present a novel PEFT method, termed RoCoFT, which finetunes selected rows and columns of model weights. Through an extensive series of experiments, we demonstrate that our method achieves competitive performance relative to other PEFT techniques, while significantly improving

both memory efficiency and training time. Furthermore, by employing kernel methods, we show that the restricted kernels generated by our approach achieve comparable accuracy to full fine-tuning kernels in kernel logistic regression tasks. This indicates that RoCoFT effectively captures the most salient features from the full parameter kernel space. Future works include combining our RoCoFT method with quantization to achieve more compressed models during finetuning. We would also like to extend the kernel approach to the study and comparison of more PEFT methods.

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

## APPENDIX A  $\Delta\mathbf{W}$ REPRESENTATION

A comparison of the $\Delta\mathbf{W}$ representations across different PEFT methods is provided in Table 8.

| Method | $\Delta\mathbf{W}$ Reparameterization | Notes |
|---|---|---|
| Intrinsic SAID | $\Delta\mathbf{W} = F(\mathbf{W}^r)$ | $\mathrm{F}: \mathbb{R}^r \to \mathbb{R}^d$, $\mathbf{W}^r \in \mathbb{R}^r$ are parameters to be optimized, and $r \ll d$. |
| LoRA | $\Delta\mathbf{W} = \mathbf{W}_{\text{down}}\mathbf{W}_{\text{up}}$ | $\mathbf{W}_{\text{down}} \in \mathbb{R}^{d\times r}$, $\mathbf{W}_{\text{up}} \in \mathbb{R}^{r\times d}$, and $r \ll \{k, d\}$. |
| KronA | $\Delta\mathbf{W} = \mathbf{W}_{\text{down}} \otimes \mathbf{W}_{\text{up}}$ | $\text{rank}(\mathbf{W}_{\text{down}} \otimes \mathbf{W}_{\text{up}}) = \text{rank}(\mathbf{W}_{\text{down}}) \times \text{rank}(\mathbf{W}_{\text{up}})$. |
| DyLoRA | $\Delta\mathbf{W} = \mathbf{W}_{\text{down}\downarrow b}\mathbf{W}_{\text{up}\downarrow b}$ | $\mathbf{W}_{\text{down}\downarrow b} = \mathbf{W}_{\text{down}}[:, b, :]$, $\mathbf{W}_{\text{up}\downarrow b} = \mathbf{W}_{\text{up}}[:, :, b]$, $b \in \{r_{\min}, \cdots, r_{\max}\}$. |
| AdaLoRA | $\Delta\mathbf{W} = \mathbf{P}\mathbf{\Lambda}\mathbf{Q}$ | $\mathbf{P}\mathbf{P}^\top = \mathbf{P}^\top\mathbf{P} \neq \mathbf{I} = \mathbf{Q}\mathbf{Q}^\top = \mathbf{Q}^\top\mathbf{Q}$, $\mathbf{\Lambda} = \text{diag}(\sigma_1, \sigma_2, \ldots, \sigma_r)$. |
| IncreLoRA | $\Delta\mathbf{W} = \mathbf{W}_{\text{down}}\Lambda W_{\text{up}}$ | $\Lambda = [\lambda_1, \lambda_2, \ldots, \lambda_r]$ with $\lambda_i$ being an arbitrary constant. |
| DeltaLoRA | $\Delta\mathbf{W} = \mathbf{W}_{\text{down}}\mathbf{W}_{\text{up}}$ | $\mathbf{W}^{(t+1)} \leftarrow \mathbf{W}^{(t)} + \left(\mathbf{W}_{\text{down}}^{(t+1)}\mathbf{W}_{\text{up}}^{(t+1)} - \mathbf{W}_{\text{down}}^{(t)}\mathbf{W}_{\text{up}}^{(t)}\right)$. |
| LoRAPrune | $\Delta\mathbf{W} = \mathbf{W}_{\text{down}}\mathbf{W}_{\text{up}} \odot \mathbf{M}$ | $\delta = (\mathbf{W} + \mathbf{W}_{\text{down}}\mathbf{W}_{\text{up}}) \odot \mathbf{M}$, $\mathbf{M} \in \{0, 1\}^{1\times G}$, $G$ is group number |
| QLoRA | $\Delta\mathbf{W} = \mathbf{W}_{\text{down}}^{\text{BF16}}\mathbf{W}_{\text{up}}^{\text{BF16}}$ | $\mathbf{Y}^{BF16} = \mathbf{X}^{BF16}\text{doubleDequant}(c_1^{FP32}, c_2^{FP8}, \mathbf{W}^{NF4}) + \mathbf{X}^{BF16}\mathbf{W}_{\text{down}}^{BF16}\mathbf{W}_{\text{down}}^{BF16}$. |
| QA-LoRA | $\Delta\mathbf{W} = \mathbf{W}_{\text{down}}\mathbf{W}_{\text{up}}$ | $\mathbf{W}_{\text{down}} \in \mathbb{R}^{d\times r}$, $\mathbf{W}_{\text{up}} \in \mathbb{R}^{r\times L}$, $L$ is the quantization group number of W. |
| LoFTQ | $\Delta\mathbf{W} = \text{SVD}(\mathbf{W} - \mathbf{Q}_t)$ | $\mathbf{Q}_t = q_N\left(\mathbf{W} - \mathbf{W}_{\text{down}}^{t-1}\mathbf{W}_{\text{up}}^{t-1}\right)$, $q_N$ is $N$-bit quantization function |
| Kernel-mix | $\Delta\mathbf{W}^h = \begin{bmatrix}\mathbf{B}_{\text{LoRA}}^h & \mathbf{B}^h\end{bmatrix}\begin{bmatrix}\mathbf{A}_{\text{LoRA}}^h \\ \mathbf{A}^h\end{bmatrix}$ | $\mathbf{B}_{\text{LoRA}}$ is shared across all heads, $\mathbf{B}_h^A$ provides rank $r$ update in each head. |
| LoRA-FA | $\Delta\mathbf{W} = \mathbf{W}_{\text{down}}\mathbf{W}_{\text{up}} = \mathbf{Q}\mathbf{R}\mathbf{W}_{\text{up}}$ | $\mathbf{W}_{\text{down}}$ is frozen, and only $\mathbf{W}_{\text{up}}$ is updated. |
| RoCoFT | $\mathbf{W} = \mathbf{W}_0 + \mathbf{R}$ 
 $\mathbf{W} = \mathbf{W}_0 + \mathbf{C}$ | $\mathbf{R}$ and $\mathbf{C}$ are restricted weight matrices such that only at most $r$ of the rows or columns are non-zero. |

Table 8: Comparison of reparameterization of various PEFT methods.

## APPENDIX B  HYPER-PARAMETERS FOR ROCOFT

The hyperparameters used in RoCoFT are provided in Table B.

| Dataset | Learning Rate | Epochs | Batch size | Dropout | Weight Decay | Warmup Steps | Learning Scheduler | Bias | Pruning | Layer Norm | Rank | Gradient Accumul. |
|---|---|---|---|---|---|---|---|---|---|---|---|---|
| CoLA | 2e-4 | 20 | 32 | 0.10 | 0.10 | 100 | cosine | True | min | 1e-05 | 3 | 0 |
| SST2 | 2e-4 | 3 | 32 | 0.10 | 0.00 | 100 | cosine | False | max | 1e-05 | 3 | 0 |
| MRPC | 2e-3 | 10 | 32 | 0.10 | 0.00 | 100 | cosine | False | random | 1e-05 | 3 | 0 |
| STS-B | 1e-3 | 10 | 32 | 0.10 | 0.00 | 100 | cosine | False | random | 1e-05 | 3 | 0 |
| QQP | 1e-4 | 2 | 32 | 0.01 | 0.00 | 100 | cosine | False | random | 1e-05 | 3 | 0 |
| MNLI | 1e-3 | 2 | 16 | 0.10 | 0.001 | 100 | cosine | False | random | 1e-05 | 3 | 0 |
| QNLI | 1e-3 | 2 | 16 | 0.10 | 0.00 | 100 | cosine | False | random | 1e-05 | 3 | 0 |
| RTE | 2e-3 | 30 | 32 | 0.10 | 0.00 | 100 | cosine | True | random | 1e-05 | 3 | 0 |
| SQuADv1.1 | 1e-4 | 4 | 16 | 0.10 | 0.00 | 100 | cosine | True | random | 1e-05 | 3 | 0 |
| SQuADv2.0 | 1e-4 | 4 | 16 | 0.10 | 0.00 | 100 | cosine | True | random | 1e-05 | 3 | 0 |
| XSum | 1e-4 | 4 | 16 | 0.10 | 0.01 | 100 | cosine | True | random | 1e-05 | 3 | 0 |
| DailyMail | 1e-4 | 4 | 16 | 0.10 | 0.01 | 100 | cosine | True | random | 1e-05 | 3 | 0 |
| BoolQ | 2e-3 | 2 | 3 | 0.10 | 0.00 | 100 | cosine | True | random | 1e-05 | 3 | 3 |
| PIQA | 2e-3 | 2 | 3 | 0.10 | 0.00 | 100 | cosine | True | random | 1e-05 | 3 | 3 |
| SIQA | 2e-3 | 2 | 3 | 0.10 | 0.00 | 100 | cosine | True | random | 1e-05 | 3 | 3 |
| Hellaswag | 2e-3 | 2 | 3 | 0.10 | 0.00 | 100 | cosine | True | random | 1e-05 | 3 | 3 |
| W.Gra. | 2e-3 | 2 | 3 | 0.10 | 0.00 | 100 | cosine | True | random | 1e-05 | 3 | 3 |
| ARCe | 2e-3 | 2 | 3 | 0.10 | 0.00 | 100 | cosine | True | random | 1e-05 | 3 | 3 |
| ARCc | 2e-3 | 4 | 3 | 0.10 | 0.00 | 100 | cosine | True | random | 1e-05 | 3 | 3 |
| OBQA | 2e-3 | 1 | 3 | 0.10 | 0.00 | 100 | cosine | True | random | 1e-05 | 3 | 3 |
| MultiArith | 1e-3 | 2 | 8 | 0.10 | 0.00 | 500 | cosine | True | random | 1e-05 | 3 | 2 |
| Gsm8k | 1e-3 | 2 | 8 | 0.10 | 0.00 | 500 | cosine | True | random | 1e-05 | 3 | 2 |
| AddSub | 1e-3 | 2 | 8 | 0.10 | 0.00 | 500 | cosine | True | random | 1e-05 | 3 | 2 |
| SingleEq | 1e-3 | 2 | 8 | 0.10 | 0.00 | 500 | cosine | True | random | 1e-05 | 3 | 2 |
| SVAMP | 1e-3 | 2 | 8 | 0.10 | 0.00 | 500 | cosine | True | random | 1e-05 | 3 | 2 |

Table 9: Hyperparameters for RoCoFT (row and column)

## APPENDIX C   ENVIRONMENTAL SETUP AND IMPLEMENTATION DETAILS

In order to implement RoCoFT, we have set up a comprehensive environment using key frameworks and tools to ensure efficient training and evaluation. We utilized PyTorch 2.4.1 as our primary deep learning framework, along with Huggingface's Transformers library version 4.44.1, which provides a wide array of pre-trained models and tokenizers, ensuring seamless integration with the RoCoFT method. To optimize the training process, we leveraged Accelerate 0.34.2, which is particularly helpful for distributed training across multiple GPUs and scaling large model deployments. This tool enabled us to efficiently manage computational resources and fine-tune the performance of large language models.

For our hardware setup, we utilized two distinct types of GPUs to optimize training based on the task requirements. For tasks like GLUE, question answering, and text summarization, we deployed NVIDIA A100 GPUs. These tasks, which are less computationally intensive compared to full LLM training, were efficiently handled by the A100s. For larger and more demanding tasks such as evaluating the performance of LLMs, we used NVIDIA H100 GPUs with 80 GB of VRAM. The H100 provided the necessary memory and computational power to handle the fine-tuning of LLMs, especially given the large model sizes and extensive data required for these tasks. This configuration allowed us to achieve significant speedups during both training and inference, while also managing memory-intensive processes with ease.

In addition to the hardware and software setup, special attention was given to the data pipeline to ensure smooth loading and processing of large datasets required for RoCoFT. Data preprocessing steps, such as tokenization and sequence padding, were handled by the Huggingface library, streamlining the preparation of input for the models. The combination of these tools and hardware resources ensured that we could efficiently implement RoCoFT across a variety of tasks while maintaining high performance and scalability.

## APPENDIX D   RoCoFT WITH RANDOM WEIGHT SELECTION

To test our hypothesis that finetuning LLMs can work as long as there are sufficient number of free parameters spread throughout the LLM model for training, we implement a version RoCoFT where instead of rows and columns, we uniformly sample entries with probability $pr$ from the weight matrices for updates and freeze the rest. Note that this method is not computationally efficient compared to updating only rows and columns and is only meant for ablation studies. From Tables 10 and 11 we can see that updating random entries in the weight matrix is competitive with all other

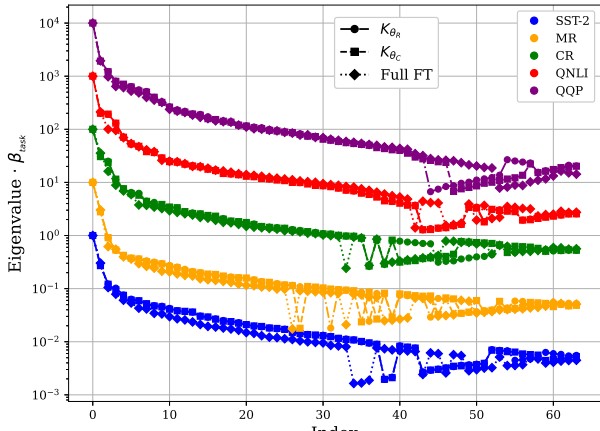

Figure 6: Eigenvalue spectrum of $\mathbf{K}_{\boldsymbol{\theta}}$, $\mathbf{K}_{\boldsymbol{\theta}_R}$, and $\mathbf{K}_{\boldsymbol{\theta}_C}$. The eigenvalues with respect to each task are scaled with $\beta_{task}$ for better representation.

PEFT methods (we use $pr = 0.1$ and $pr = 0.01$ in these experiments). This gives further evidence that most good features are already acquired during pretraining and little learning is required during the finetuning stage.

| LM | # TTPs | CoLA | SST2 | MRPC | STS-B | QQP | MNLI | QNLI | RTE |
|---|---|---|---|---|---|---|---|---|---|
| Roberta$_{Base}$ | 12.4M | 63.15 | 94.96 | 88.03/89.18 | 90.57/90.07 | 89.29/86.94 | 87.22 | 92.60 | 80.01 |
| Roberta$_{Large}$ | 35.5M | 65.32 | 96.59 | 90.93/92.03 | 92.10/92.05 | 90.97/86.78 | 90.89 | 95.06 | 87.91 |

Table 10: RoBERTa models performance on GLUE tasks using 10% random sampling of trainable parameters from each weight matrix ($pr = 0.1$).

| LLM | # TTPs | BoolQ | PIQA | SIQA | H.Sw. | W.Gra. | ARCe | ARCc | OBQA | M.Ar. | G.8K | A.S. | S.eEq | S.MP |
|---|---|---|---|---|---|---|---|---|---|---|---|---|---|---|
| BLOOMz$_{7B}$ | 70.4M | 65.76 | 74.62 | 73.50 | 56.39 | 72.11 | 72.89 | 56.88 | 72.43 | 79.78 | 71.11 | 70.76 | 70.91 | 54.37 |
| GPT-J$_{6B}$ | 60.3M | 65.75 | 68.63 | 69.12 | 45.50 | 66.47 | 64.99 | 46.91 | 65.37 | 89.34 | 72.62 | 80.64 | 82.14 | 55.90 |
| LLaMA2$_{7B}$ | 71.2M | 69.30 | 80.12 | 77.95 | 89.40 | 76.52 | 76.57 | 60.62 | 76.92 | 90.46 | 77.32 | 86.13 | 82.49 | 60.72 |
| LLaMA2$_{13B}$ | 129.8M | 71.44 | 83.37 | 79.32 | 91.95 | 83.32 | 83.99 | 66.92 | 81.32 | 91.49 | 80.04 | 87.71 | 87.64 | 66.83 |

Table 11: Accuracy comparison of commonsense and mathematical reasoning performance across different datasets using LLMs, using 1% random sampling of total trainable model parameters from each weight matrix ($pr = 0.01$).

## APPENDIX E    ADDITIONAL NEURAL TANGENT KERNEL RESULTS

Here we include additional results on our Neural Tangent Kernel experiments. Figure 6 shows the eigenvalue distribution of the full kernel $\mathbf{K}_{\boldsymbol{\theta}}$, 1-row kernel $\mathbf{K}_{\boldsymbol{\theta}_R}$ and 1-column kernel $\mathbf{K}_{\boldsymbol{\theta}_C}$ on different datasets. The eigenvalues are rescaled per dataset and we can see the eigenvalue distributions are very similar for the three NTK kernels. Table 12 shows the $\ell_1$ and $\ell_2$ norm difference between the kernel matrices of the 64-shot tasks, and the results are largely similar to the 16-shot results. The difference is mostly within 5-15%, but with smaller standard deviation than the 16-shot results over 5 random seeds. In Figure 7, we include a few more visualizations of the kernel matrices for the 16-shot tasks. We can see the three type of NTK matrices show very similar patterns across all tasks.

## APPENDIX F    DATASET DESCRIPTION

The datasets used in this study are listed in Table 13 and Table 14.

| 64-shot (single) | SST-2 | SST-5 | MR | CR | MPQA | Subj | TREC |
|---|---|---|---|---|---|---|---|
| $\mathbf{K}_{\theta_R}, p=1$ | 0.091(0.007) | 0.084(0.002) | 0.067(0.005) | 0.084(0.005) | 0.126(0.014) | 0.061(0.002) | 0.184(0.003) |
| $\mathbf{K}_{\theta_R}, p=2$ | 0.126(0.012) | 0.113(0.002) | 0.100(0.015) | 0.115(0.013) | 0.176(0.025) | 0.076(0.008) | 0.202(0.004) |
| $\mathbf{K}_{\theta_C}, p=1$ | 0.088(0.007) | 0.079(0.002) | 0.064(0.005) | 0.080(0.005) | 0.125(0.015) | 0.055(0.002) | 0.169(0.003) |
| $\mathbf{K}_{\theta_C}, p=2$ | 0.124(0.012) | 0.108(0.003) | 0.098(0.014) | 0.110(0.011) | 0.178(0.026) | 0.071(0.004) | 0.191(0.004) |

| 64-shot (pair) | MNLI | SNLI | QNLI | RTE | MRPC | QQP | |
|---|---|---|---|---|---|---|---|
| $\mathbf{K}_{\theta_R}, p=1$ | 0.181(0.012) | 0.205(0.013) | 0.074(0.013) | 0.128(0.004) | 0.073(0.009) | 0.049(0.007) | |
| $\mathbf{K}_{\theta_R}, p=2$ | 0.251(0.037) | 0.259(0.033) | 0.179(0.069) | 0.180(0.011) | 0.093(0.004) | 0.099(0.065) | |
| $\mathbf{K}_{\theta_C}, p=1$ | 0.179(0.013) | 0.200(0.014) | 0.071(0.013) | 0.125(0.005) | 0.073(0.003) | 0.048(0.007) | |
| $\mathbf{K}_{\theta_C}, p=2$ | 0.254(0.040) | 0.257(0.034) | 0.172(0.065) | 0.186(0.013) | 0.093(0.004) | 0.099(0.067) | |

Table 12: Relative difference in kernels (compared to full parameter $\mathbf{K}_\theta$) on single-sentence and sentence-pair tasks for 64-shot tasks

| Dataset | Domain | Train | Test |
|---|---|---|---|
| MultiArith | Math | – | 600 |
| AddSub | Math | – | 395 |
| GSM8K | Math | 8.8K | 1,319 |
| AQuA | Math | 100K | 254 |
| SingleEq | Math | – | 508 |
| SVAMP | Math | – | 1,000 |
| BoolQ | CS | 9.4K | 3,270 |
| PIQA | CS | 16.1K | 1,830 |
| SIQA | CS | 33.4K | 1,954 |
| HellaSwag | CS | 39.9K | 10,042 |
| WinoGrande | CS | 63.2K | 1,267 |
| ARC-e | CS | 1.1K | 2,376 |
| ARC-c | CS | 2.3K | 1,172 |
| OBQA | CS | 5.0K | 500 |

Table 13: Overview of Datasets for Mathematical and Commonsense Reasoning

| Dataset | Train | Validation | Test |
|---|---|---|---|
| SQuAD v1.1 | 87.6k | 10.6k | - |
| SQuAD v2.0 | 130k | 11.9k | - |
| XSum | 204k | 11.3k | 11.3k |
| DailyMail | 287k | 13.4k | 11.5k |
| CoLA | 8.55k | 1.04k | 1.06k |
| SST2 | 67.3k | 872 | 1.82k |
| MRPC | 3.67k | 408 | 1.73k |
| STS-B | 5.75k | 1.5k | 1.38k |
| QQP | 364k | 40.4k | 391k |
| MNLI | 393k | 9.8k | 9.8k |
| QNLI | 105k | 5.46k | 5.46k |
| RTE | 2.49k | 277 | 3k |

Table 14: Summary of Datasets for GLUE, Question Answering, and Text Summarization

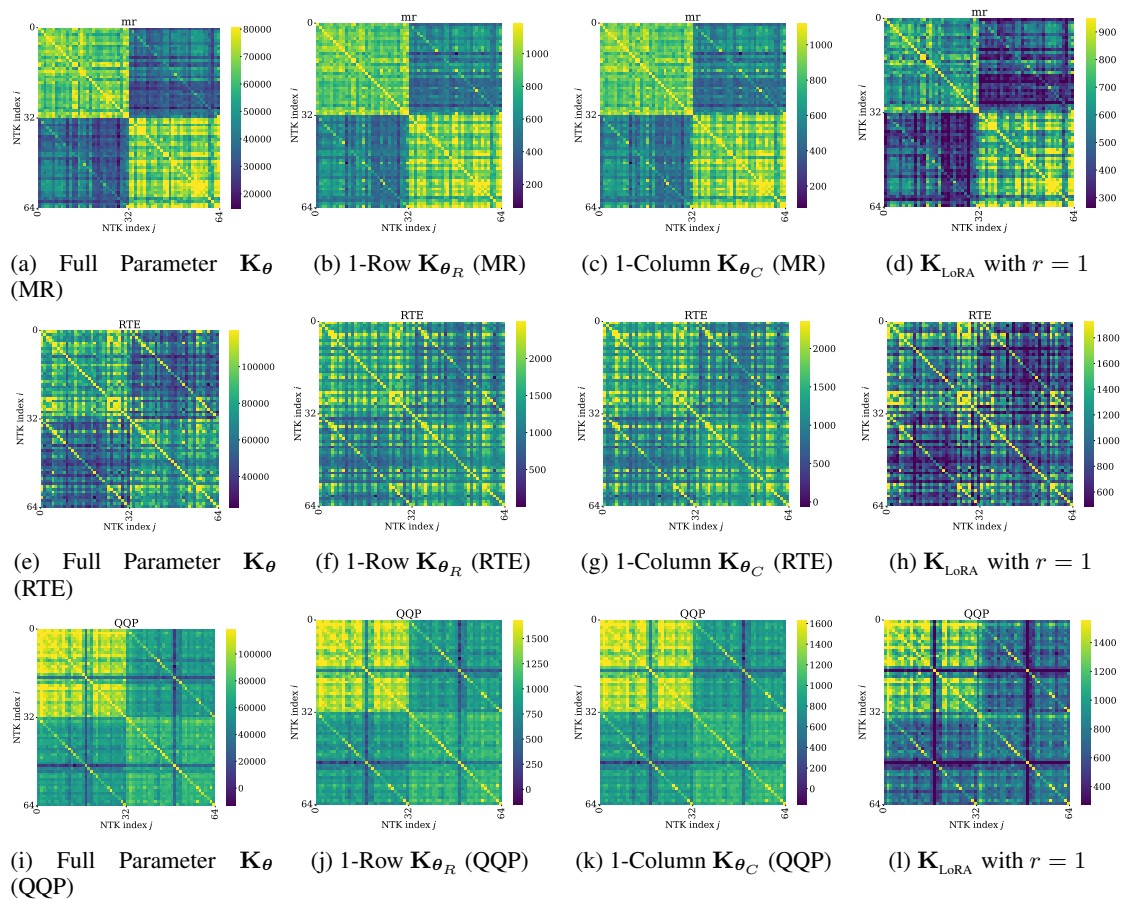

(a) Full Parameter $\mathbf{K}_{\boldsymbol{\theta}}$ (MR)   (b) 1-Row $\mathbf{K}_{\boldsymbol{\theta}_R}$ (MR)   (c) 1-Column $\mathbf{K}_{\boldsymbol{\theta}_C}$ (MR)   (d) $\mathbf{K}_{\text{LoRA}}$ with $r = 1$

(e) Full Parameter $\mathbf{K}_{\boldsymbol{\theta}}$ (RTE)   (f) 1-Row $\mathbf{K}_{\boldsymbol{\theta}_R}$ (RTE)   (g) 1-Column $\mathbf{K}_{\boldsymbol{\theta}_C}$ (RTE)   (h) $\mathbf{K}_{\text{LoRA}}$ with $r = 1$

(i) Full Parameter $\mathbf{K}_{\boldsymbol{\theta}}$ (QQP)   (j) 1-Row $\mathbf{K}_{\boldsymbol{\theta}_R}$ (QQP)   (k) 1-Column $\mathbf{K}_{\boldsymbol{\theta}_C}$ (QQP)   (l) $\mathbf{K}_{\text{LoRA}}$ with $r = 1$

Figure 7: Neural Tangent Kernels on 16-shot training data for different tasks

## APPENDIX G   EVALUATION METRICS

We employ specific evaluation metrics tailored to each task within the GLUE benchmark suite (Wang et al., 2018) to assess the performance of our models comprehensively.

For the **Corpus of Linguistic Acceptability (CoLA)** task, we use the *Matthews Correlation Coefficient* (MCC) as the evaluation metric. MCC is suitable for binary classification tasks, especially with imbalanced datasets, as it takes into account true positives (TP), true negatives (TN), false positives (FP), and false negatives (FN):

$$\text{MCC} = \frac{\text{TP} \times \text{TN} - \text{FP} \times \text{FN}}{\sqrt{(\text{TP} + \text{FP})(\text{TP} + \text{FN})(\text{TN} + \text{FP})(\text{TN} + \text{FN})}}. \tag{4}$$

For the **Microsoft Research Paraphrase Corpus (MRPC)** and **Quora Question Pairs (QQP)** tasks, which evaluate the model's ability to determine semantic equivalence between sentence pairs, we use both *Accuracy* and *F1 Score* as evaluation metrics. Accuracy measures the proportion of correctly identified paraphrase pairs, while the F1 score balances precision and recall:

$$\text{Accuracy} = \frac{\text{TP} + \text{TN}}{\text{TP} + \text{TN} + \text{FP} + \text{FN}}, \tag{5}$$

$$\text{F1} = 2 \times \frac{\text{Precision} \times \text{Recall}}{\text{Precision} + \text{Recall}}, \tag{6}$$

where precision and recall are defined as:

$$\text{Precision} = \frac{\text{TP}}{\text{TP} + \text{FP}}, \quad \text{Recall} = \frac{\text{TP}}{\text{TP} + \text{FN}}. \tag{7}$$

For the **Multi-Genre Natural Language Inference (MNLI)** task, which involves classifying sentence pairs into *entailment*, *contradiction*, or *neutral*, we report the *Average Matched Accuracy*. This metric measures the model's accuracy on the matched validation set (in-domain data), reflecting its ability to generalize across different genres.

For the **Semantic Textual Similarity Benchmark (STS-B)** task, which requires predicting the degree of semantic similarity between sentence pairs, we use both the *Pearson* and *Spearman* correlation coefficients. These metrics evaluate the linear and rank-order relationships between the predicted scores ($x_i$) and the ground-truth scores ($y_i$), respectively:

$$\text{Pearson's } r = \frac{\sum_{i=1}^{n}(x_i - \bar{x})(y_i - \bar{y})}{\sqrt{\sum_{i=1}^{n}(x_i - \bar{x})^2}\sqrt{\sum_{i=1}^{n}(y_i - \bar{y})^2}}, \tag{8}$$

$$\text{Spearman's } \rho = 1 - \frac{6\sum_{i=1}^{n} d_i^2}{n(n^2 - 1)}, \tag{9}$$

where $\bar{x}$ and $\bar{y}$ are the means of the predicted and ground-truth scores, $d_i$ is the difference between the ranks of $x_i$ and $y_i$, and $n$ is the number of data points.

These evaluation metrics provide a comprehensive assessment of our models across diverse linguistic tasks, enabling us to measure both classification accuracy and the ability to capture semantic nuances.

