# OpenReview forum: "RoCoFT: Efficient Finetuning of Large Language Models with Row-Column Updates"
_ICLR.cc/2025/Conference — ICLR 2025 Conference Withdrawn Submission_

### Official Review · Reviewer_ZG6N · 2024-10-16

**Soundness:** 2
**Presentation:** 3
**Contribution:** 3
**Rating:** 6
**Confidence:** 3

**Summary:**

The authors propose a novel method named RoCoFT for parameter-efficient fine-tuning (PEFT). RoCoFT updates only a few rows or columns of the trained parameter matrices, achieving even lower complexity compared to existing PEFT methods. The effectiveness of RoCoFT is supported by neural tangent kernel (NTK) theory, as demonstrated by the authors. The empirical performance of RoCoFT is extensively evaluated on several benchmarks and compared with a large number of baselines.

**Strengths:**

1. The method is simple, straightforward, yet effective. The presentation is clear and easy to follow.

2. The performance comparison with baselines is extensive. Besides, the learnable parameters in RoCoFT are much less than existing methods, which is very useful.

3. As shown in ablation studies, the strategy of choosing rows and columns is robust and does not need much tuning.

**Weaknesses:**

1. The NTK analysis in Section 5 is not complete. The results in Tables 5 and 6 only include comparisons between RoCoFT, FT, and the pre-trained weights. However, if other methods, such as LoRA, also have a kernel that is empirically close to the full-parameter kernel, it becomes unclear why RoCoFT can achieve performance improvements over them. Similar experiments on other baselines should also be included.
2. Further explanation should be provided on why the few-shot learning performance is used as a downstream task for kernel comparison in Tables 5 and 6. Why are the performances in Tables 1, 2, and 3 not used for kernel comparison?
3. The empirical improvement in memory costs in Figure 2 and training time costs in Figure 3 appears marginal, which is inconsistent with the large improvement suggested by Table 4. Please provide a detailed explanation.

**Questions:**

1. How are the two values in the "Avg." column computed in Table 1?

2. Can the row update and column update be used simultaneously? It seems to me that this simple strategy allows for more flexibility and enhanced performance.

---

> ### Author Response · Authors · 2024-11-22
> **Response to weakness points: updates with new numerical experiments.**
>
> We thank the reviewer for detailed feedback recommendations  which enriched our work.
> ##  W1. NTK analysis for LoRA
> Following your suggestion we have tried running NTK on LoRA with rank 1 on a few datasets. The kernel regression results on 16-shot learning are as follows (compared to Table 5):
>
> | Dataset | SST-2 | MR| CR| QNLI| RTE| QQP |
> |---------|------------|---------|-----------|--------|-----------------|--------|
> | LoRA   |   88.5(0.7)    | 84.5(1.4)  | 93.2(0.5)     | 59.9(3.0)  |    58.8(4.7)       |  58.2(2.6) |
>
> We have also updated the  Figures 4 and 7 of our paper on corresponding NTK plots for these datasets. They look similar to the NTK for full parameter set but are not visually as close to the NTK for row/column parameters. But in terms of relative l1/l2 distance to the NTK full parameter set (compared to Table 6) the NTK for LoRA is also close:
>
> | Metric  | SST2           | MR             | CR             | QNLI           | RTE            | QQP            |
> |---|----------------|----------------|----------------|----------------|----------------|----------------|
> | p=1     | 0.090 (0.022)  | 0.086 (0.021)  | 0.108 (0.024)  | 0.077 (0.017)  | 0.119 (0.027)  | 0.084 (0.021)  |
> | p=2     | 0.108 (0.028)  | 0.102 (0.027)  | 0.132 (0.036)  | 0.103 (0.025)  | 0.150 (0.039)  | 0.106 (0.027)  |
>
> However, our motivation for showing the NTKs for full parameter set and row/column parameters are close is to provide evidence to explain why FT and RoCoFT can have similar performance. We do not intend to use the similarity to the NTK of the full parameter set as a way to rank the performance of PEFT methods, as there are several subtleties involved, including how well the NTK kernel regression can approximate each of the PEFT method. For example, for LoRA since there are new adaptor variables involved, unlike the NTK for FT or RoCoFT, the corresponding NTK for LoRA depends on how those variables are initialized (we use default initialization in the huggingface PEFT library in the above experiemnts). The closeness of the NTKs of these PEFT methods to the NTK of full parameter set just suggest the corresponding PEFT method could have performance close to full finetuning. But it would be indeed an interesting future work to use NTKs to perform a more refined analysis of different PEFT methods.
> ## W3. Empirical improvement in memory cost
> The apparent discrepancy between the marginal improvements in memory costs (Figure 2) and training time costs (Figure 3) compared to the larger improvements suggested in Table 4 arises because Figures 2 and 3 reflect experiments conducted with the Adam optimizer, which inherently doubles the memory requirement due to maintaining additional states (e.g., moment estimates) for each parameter. This effect reduces the relative advantage of RoCoFT, as the optimizer memory dominates the overall cost. In contrast, Table 4 highlights the theoretical memory and computational efficiency specific to RoCoFT's architecture, independent of optimizer overhead, demonstrating the substantial savings achieved by our method in terms of trainable parameters and memory efficiency. We will clarify this distinction in the revised paper.
>
> ## W2. Few-shot learning for kernel comparison
> This is mainly due to high computational costs of computing NTK matrices, which scales as $N^2$ where $N$
>  is the number of training examples. As we are trying to understand why FT and RoCoFT can have similar performance using NTK analysis, we believe performing the analysis on a representative subset of the datasets is sufficient. Also the results for K=64 shot are already fairly close to finetuning using the full training set on many datasets.

---

> ### Author Response · Authors · 2024-11-22
> **Response to questions**
>
> ## Q1.  "Avg." column of Table 1
> The two values in the "Avg." column of Table 1, presented as 'a' or 'a/b', are computed as separate averages based on the two sets of metrics reported in the table. The first value, 'a', represents the average of the first metric (e.g., MCC, Accuracy, etc.) across all tasks, while the second value, 'b', represents the average of the second metric (e.g., F1 score or another secondary metric) across all tasks. We understand that this column was  confusing therefore we removed it in the revised manuscript. Additionally, we edited all captions to reflect what metrics  'a/b' are.
>
> ## Q2. Simultaneous Row-Column updates
> Thank you for this suggestion. While it is possible to update both rows and columns simultaneously, doing so presents practical challenges in our current implementation. Currently, we split each weight matrix into two parts: trainable and non-trainable, based on either rows or columns. Maintaining this partitioning while updating both rows and columns introduces overlap in the trainable parameters, which complicates the setup and management of the updates.
>
> One approach we explored was to use masking to manage simultaneous updates, as described in Appendix D ("RoCoFT with Random Weight Selection"). However, this method did not achieve meaningful memory reduction because the masking operation requires additional memory allocation and storage for the mask itself, akin to the inefficiencies observed with random selection strategies.
>
> In future work, we aim to devise a more efficient strategy for jointly controlling and updating both rows and columns, ensuring minimal overlap and optimal memory efficiency while maintaining the performance benefits of this combined approach.

---

> ### Author Response · Authors · 2024-11-30
>
> Dear reviewer ZG6N,
>
> We are grateful for your constructive feedback on NTK regression section, which led us to  further  investigate  the comparison between LoRA, RoCoFT and  full fine-tuning through the perspective of the NTK. We would be happy to address any remaining concerns you may have regarding the revised manuscript and the new experimental results. Please feel free to provide further comments or suggestions, and we will make every effort to incorporate them. We look forward to your feedback.
>
> Sincerely,
>
> Authors

---

> ### Author Response · Authors · 2024-12-02
>
> Dear reviewer ZG6N,
>
> Kindly,  please let us know of  any remaining concerns about our paper. We will be happy to address them.
>
> Sincerely,
>
> Authors.

---

> ### Comment · Reviewer_ZG6N · 2024-12-03
>
> Thanks for your clarification, which addressed most of my concerns. I will raise my score to 6.
>
> Just an additional comment: I think there should be some deep connections between RoCoFT and LoRA with rank = 1. Essentially, let B and A in LoRA be of dimensions $d\times 1$ and $1\times d$, and restrict A to be non-learnable with only one value in one position and zero in the other positions. Then, this type of restricted LoRA incremental matrix should reduce to your column-update scheme. A similar argument should hold for the row-update scheme. Therefore, I believe the RoCoFT method may also be explained from this perspective and could be potentially explored in future work.

---

### Official Review · Reviewer_34VR · 2024-10-29

**Soundness:** 2
**Presentation:** 3
**Contribution:** 2
**Rating:** 6
**Confidence:** 4

**Summary:**

The authors proposed a simple fine tuning method for LLMs that updates only a few columns/rows in the base model. NTK regression-based analysis proposed to explain why single row/column updates work and extensive experiments were conducted to evaluate the method on diverse language tasks.

**Strengths:**

- The method is very simple but shows prominent results for some datasets
- The method was evaluated on large and diverse number of datasets
- Applying NTK regression to get explanation for why the method works - looks interesting

**Weaknesses:**

- Limited novelty of the proposed method: the authors propose to update a few columns/rows in the base model and exploit the existing NTK regression method to explain it.
- I don’t understand how the results in Table 5 are consistent with Table 1 so we can explain why the method works with NTK regression. In Table 5 the proposed method performs worse than FT while in Table 1 it is not the case.
- In-place updates disable the behavior of the model as an adaptor. This is a trade-off that should be discussed while presenting 0 additional parameters
- Missing explanation / intuition why the method fails on some datasets, e.g. MNLI, QNLI, RTE in Table 1.
- Missing additional recent LoRA-style baselines with low number of trainable parameters, e.g. [1-3]
- The efficiency gains are not significant compared to other LoRA-style methods, also it is not interesting since the number of trainable parameters is small for adapter-like methods.


[1] Bałazy, Klaudia, et al. "LoRA-XS: Low-Rank Adaptation with Extremely Small Number of Parameters." arXiv preprint arXiv:2405.17604 (2024).

[2] Kopiczko, Dawid J., Tijmen Blankevoort, and Yuki M. Asano. "Vera: Vector-based random matrix adaptation." arXiv preprint arXiv:2310.11454 (2023).

[3] Zhang, Longteng, et al. "Lora-fa: Memory-efficient low-rank adaptation for large language models fine-tuning." arXiv preprint arXiv:2308.03303 (2023).

**Questions:**

- I would like to see an experiment where no weights are updated in the pretrained model and only classification head is trained and how the obtained accuracy differs from the single-column/row adaptations.
- It is not clear from the Sec.4 if the setup of baselines in terms of hyper parameters is the same as of the proposed method. I’m concerned that the small differences in the evaluation between the proposed method and baselines stems from setup differences.

---

> ### Author Response · Authors · 2024-11-22
> **Response to the weakness points**
>
> We thank the  reviewer for raising insightful points. Below, we address the concerns thoroughly.
>
> ## W1.  Limited novelty
> The novelty of  our paper is twofold. First, to the best of our knowledge, the row-column update of the weight matrices despite being starkly low-complexity and   showing competitive accuracies, is not proposed as a method in previous papers. Moreover, explaining the effectivity of fine-tuning method by means of NTK regression is at its infancy in the literature. We believe the NTK regression, when  finetuning is lazy,  is a powerful tool  for analyzing the learning dynamics of  finetuning.
>
> ## W2. Consistency of Table 5 with Table 1
> We are sorry for the confusion. There are actually several differences between Table 1 and Table 5. Table 1 is the comparison of our method RoCoFT with other PEFT methods and FT on standard benchmarks, while Table 5 is a comparison of FT against kernel regression using NTKs derived from full parameter set and the row/column parameter set under a few-shot learning setting. Table 1 finetunes on the whole training set while Table 5 uses few-shot learning because computing the NTKs on large training set is expensive. Also, kernel regression using NTKs on the full parameter set and the row/column parameter set in Table 5 are approximations to FT and our RoCoFT methods. The kernel regression performance is usually a little lower than actual finetuning with backpropagation. We are not advocating using kernel regression with NTKs to replace finetuning with backprop (whether FT or RoCoFT). We are just using kernel regression with NTKs to provide an independent view on why the performance of FT and RoCoFT can be close.
>
> ## W3. Behavior of the model as an adaptor with In-place updates
> Actually in-place updates do not disable the use of our method as adaptors. Once RoCoFT is done we can compute the difference of the row/column updates with the corresponding original row/column values in the pretrained model, and store these differences as adaptors. This is the same as LoRA except LoRA express the adaptor as a difference added to the original weight matrix before finetuning while RoCoFT needs to do some postprocessing to obtain the corresponding adaptors.
>
> ## W4. Performance on  MNLI, QNLI, RTE datasets
> For the question on MNLI, QNLI, RTE in Table 1, perhaps we don't understand the question fully. The results of RoCoFT on these 3 datasets are competitive with the rest of the PEFT methods even when they are not the best among all the methods. The results for 1 row/column are weaker due to limited number of parameters, but the 3 row/column results are promising.
>
> ## W5. Additional baseline methods
> We thank you for the valuable feedback. We recognize the importance of comparing RoCoFT directly with sparse  recent LoRA-style baselines.
>
> In response to your suggestion, we have included a new comparison with recent works, including sparse fine-tuning methods. Additionally, we will incorporate further baseline comparisons with well-established sparse fine-tuning methods in our experiments to strengthen the evaluation.
>
> | Dataset | LoRA-XS[1] | Vera[2] | LoRAFA[3] | SFT[4] | Diff Pruning[5] | FSM[6] | RoCoFT (row) | RoCoFT (column) |
> |---------|------------|---------|-----------|--------|-----------------|--------|--------------|-----------------|
> | SST2    | 93.19      | 93.89   | 93.65     | 94.28  | 93.77           | 94.11  | 94.92        | 94.69           |
> | CoLA    | 58.49      | 60.35   | 60.49     | 64.45  | 62.45           | 62.77  | 63.53        | 62.95           |
> | MNLI    | 85.34      | 85.64   | 86.11     | 86.64  | 85.32           | 85.85  | 86.73        | 86.76           |
> | QNLI    | 90.42      | 90.22   | 91.42     | 92.11  | 92.14           | 91.81  | 92.12        | 91.89           |
>
>
>
> [1] LoRA-XS: Low-Rank Adaptation with Extremely Small Number of Parameters.
> [2] Vera: Vector-based random matrix adaptation.
> [3] Lora-fa: Memory-efficient low-rank adaptation for large language models fine-tuning.
> [4] Scaling Sparse Fine-Tuning to Large Language Models
> [5]  Diff Pruning: Parameter-Efficient Transfer Learning with Diff Pruning
> [6] The Lottery Ticket Hypothesis: Finding Sparse, Trainable Neural Networks
>
> ## W6. Efficiency gains
> Thank you for your observation. While it may seem that the efficiency gains compared to LoRA-style methods are modest, our results demonstrate that RoCoFT consistently outperforms LoRA-type methods in terms of accuracy, parameter efficiency, memory usage, and training time, as shown in the results section.
>
> The novelty of RoCoFT lies in its simplicity and its approach of directly modifying existing model parameters without introducing additional adapters or external modules. Unlike adapter-based methods, which require extra trainable parameters and introduce memory overhead, RoCoFT avoids these complexities while still achieving competitive or superior performance.

---

> ### Author Response · Authors · 2024-11-22
> **Response to questions**
>
> ## Q1. Performance with updating only classification head
> Thank you for your suggestion. We conducted an additional experiment where no weights in the pretrained model were updated, and only the classification head was trained. We compared this with the single-column and single-row adaptations of RoCoFT. The results for the SST-2 and MNLI datasets are as follows:
> | Dataset | Classification Head Only | Single-Column Adaptation | Single-Row Adaptation |
> |---------|---------------------------|---------------------------|-----------------------|
> | SST-2   | 88.29%                   | 93.88%                   | 94.06%               |
> | MNLI    | 80.82%                   | 85.35%                   | 85.23% |
>
> ## Q2. Hyper parameters
>  Thank you for pointing this out. To ensure consistency in evaluating baselines and our proposed method, we have followed the experimental setups described in Xu et al. (2023)[1] and Zhang et al. (2023a)[2], as referenced in Section 4. Additionally, we have provided detailed information about our environmental setup and implementation details in Appendix C.
>
> To address any potential ambiguity, we  clarified this in the revised version of the paper by explicitly stating that the hyperparameter settings for the baselines were aligned with those in these prior works, and we ensured that any setup differences are clearly outlined.
>
> [1] Parameter-efficient fine-tuning methods for pretrained language models: A critical review and assessment
>
> [2] Adaptive budget allocation for parameter-efficient fine-tuning.

---

> > ### Comment · Reviewer_34VR · 2024-11-25
> >
> > Dear authors, thank you for your efforts to provide additional clarifications.
> >
> > **W2. Consistency of Table 5 with Table 1**
> > Thank you for your clarification, I would suggest clarifying it further in the manuscript rather than addressing Malladi et al. (2023) work.
> >
> > **W3. The behavior of the model as an adaptor with In-place updates**
> > Please add it to the main text.
> >
> > **W4. Performance on MNLI, QNLI, and RTE datasets**
> > I'm sorry for the using wrong terminology. I meant why the method's performance is not better on these datasets like on others.
> >
> > **W5. Additional baseline methods**
> > Thank you for the additional results. I would suggest including these results and results on other datasets from GLUE benchmark also in your revised manuscript.
> >
> > **W6. Efficiency gains**
> >
> > I agree that simplicity is the key component of your method. However, the main contribution of a new adaptor method should be, in my view, its accuracy.
> > By considering new methods that compress the base model and/or train small adaptors, the memory efficiency of the proposed method is a less desired property.
> > Moreover, it is hard to see any significance in the accuracy gains of the proposed method compared to the baselines.
> >
> > While I see this method as simple and competitive with the baselines, it is hard to see additional novelties that should be published in this top-tier conference. In addition, while the method is evaluated on a diverse set of datasets, more recent baselines should be included with complete comparison (like all datasets from GLUE).
> >
> > I also don't any changes in the manuscript.
> >
> > I prefer to preserve my score.

---

> ### Author Response · Authors · 2024-11-29
>
> Dear reviewer 34VR
>
> Thank you for raising these important points.
>
> $\bullet$ We do not make major changes to the main manuscript yet due to page limit and the different additional results
> requested by the four reviewers. Some of the additional results were already added to the appendix. We will
> include the clarifications and replies on additional results in these discussions into the main paper once these
> discussions are finalized.
>
> $\bullet$ As for the question of the importance of accuracy VS memory efficiency in the research of PEFT methods,
> we believe BOTH are very important. We think of the advancements of PEFT methods as pushing the Pareto
> frontier for accuracy and memmory efficiency for finetuning LLMs. Therefore when we look at the results like
> those in Tables 1-3, we don’t just consider the accuracy numbers, but also the number of trainable parameters.
> Also if we really care about accuracy, the best way to improve accuracy is to use more trainable parameters
> in finetuning (e.g. increasing rank from 1 to 3), or a pretrained model of larger size. Having a more memory-
> efficient method will enable us to employ larger models and more trainable parameters. And as for methods
> that compress or quantize the base models, we believe they are independent methods to promote memory
> efficiency that can be used in conjunction with PEFT methods(e.g. QLoRA). The existence of compression or
> quantization methods for LLMs does not make the research on memory efficiency in PEFT methods redundant.
>
> $\bullet$  We are confident that in addtion to posposing the RoCoFT method,  explaining the effectivity of fine-tuning method by means of NTK regression is the novelty of our paper. Following the insightful comments of reviewers, we added the NTK regression results on LoRA to the revised paper which  shows that the NTK for LoRA with r = 1 is not as close
> as the NTK for row/column parameters to the full parameter kernel. This view of PEFT as a kernel machine has an important  impact on PEFT research.
>
> Sincerely,
>
> Authors

---

> > ### Comment · Reviewer_34VR · 2024-12-02
> >
> > Thank you for your clarifications. I raise my score.

---

> ### Author Response · Authors · 2024-12-02
>
> Thank you for the insightful comments and raising your evaluation score.

---

### Official Review · Reviewer_cVGh · 2024-11-01

**Soundness:** 1
**Presentation:** 1
**Contribution:** 1
**Rating:** 3
**Confidence:** 4

**Summary:**

In this paper, the authors address the challenge of efficiently adapting a large language model to a new task. This problem, known as Parameter Efficient Fine Tuning (PEFT), has gained significant attention in recent years following Lora's success. The main observation in this paper is that training only a small subset of rows or columns of the original weight matrices is sufficient for attaining good performance on the new task. This means fine-tuning could be performed by updating a few parameters with no memory overhead (as required with Lora-style methods). This type of fine-tuning is evaluated on multiple datasets and using several base models. The results demonstrate that this approach is competitive with leading baselines.

**Strengths:**

The authors study an important problem in LLMs.

The method is relatively efficient and lightweight.

The evaluation covers multiple transfer tasks and several base models.

They provide an NTK based empirical evaluation that aims to explain the observed phenomenon.

**Weaknesses:**

The paper is not well written, multiple parts are not clear, and there are many typos.

In essence, the method presented in this paper was already presented in another paper [1].
In fact, in [1] the authors wrote:

“ We randomly sampled the same amount of parameters as in BitFit from the entire model, and fine-tuned only them (“rand uniform” line in Table 3). The results are substantially worse across all tasks; similar patterns are observed when the random parameters are sampled as complete rows/columns in the parameter matrices (“rand row/col” line in Table 3). “

Which basically indicates that the authors in [1] have already evaluated the procedure detailed in this paper and concluded that updating the bias terms (also known as fitbit) is better.
The results by the authors demonstrate comparable results between single row/column updates and fitbit. In contrast, the authors in [1] demonstrated that row/column update does not work well in some datasets. Can the authors explain why there is a performance gap between the evaluation in [1] and what is reported in this paper?


Another problem, is that proper credit is not given to [1], which were the first to propose using row/column updates.


Even when ignoring the fact that this idea was already presented in [1], I can still see value in providing new insights about this row/column optimization scheme. But I don’t see the paper providing such new insights in its current form. The row/column selection strategy is pretty standard, and the evaluated selection schemes work as well as random selection.



[1] Ben Zaken et al. BitFit: Simple Parameter-efficient Fine-tuning for Transformer-based Masked Language-models.

The NTK perspective is also unclear; since it is primarily empirical, I don’t see what new intuition is gained from these evaluations. It is intuitive that changing only one raw/column from the entire matrix won’t change the NTK much, but also that a few steps of fine-tuning won’t. If anything, the authors should have also compared these kernels to the original kernel (before fine-tuning). If they are all similar to the original one (before fine-tuning), then I don’t understand what we gain from this insight.

In terms of presentation, the paper needs significant improvement. Currently, the results are presented without providing a clear explanation of the “method.” Specifically, the scheme for the selection of rows and columns is only described in the results section.

**Questions:**

In the results, the authors detail methods termed 1-row and 3-row without explaining what those are.

Also, regarding the number of parameters, it seems that the 3-row uses 5 times the number of parameters as the 1-row. So, is it three rows vs. one? Something does not make sense here.

Why are the methods presented in Table 1 not included in all other tables?
For example, why isn’t Fitbit (which is the most related paper to this one) not included in Tables 2+3+4, figure 2+3?





Multiple typos:

 "paradiagm" -> "paradigm".
"mermory" -> "memory".
"signficant"  -> "significant".
"tranformer"  -> "transformer".

“computation-efficient”  -> “computationally efficient.”
In the abstract “our kernel…are numerically”-> should be *is* numerically

Several abbreviations are mentioned in the abstract without introducing what they mean: RoCoFT, PEFT..

No intuition about the selection is provided in the abstract.

Many times, two numbers are presented without explaining what they mean, for example, in line 203 85.65/90.61?! And in many cases, in the tables. This is not clear, even from the caption, which tries to explain what they mean.

In some cases, as shown in Table 1, FT is substantially worse than many low-rank methods, for example, in RTE. Doesn’t this suggest that there is severe overfitting?

In the optimal rank evaluation, the performance of the RoCoFT method is not consistent with the results of the same method presented in Table 1 (for this data, SST2).


Overall, I would recommend the authors rewrite the paper as an “insight paper”, which provides empirical evaluations that support a phenomenon, rather than a “method paper”. It would also be valuable to look into the dedicated scheme for selecting the rows/columns.

---

> ### Author Response · Authors · 2024-11-23
> **Response to weakness points 1-3**
>
> We sincerely appreciate the time and effort the review has dedicated to evaluating our submission. We  would be happy to discuss any additional concerns the reviewer may have.
>
> ## W1. Typos and clarity
> Thank you for  raising these points. In the revised manuscript, we corrected the typos and clarified the points raised by the reviewers. The changes to the text is  marked with "..." in the response to the reviewers.
>
> ## W2. BitFit
> Thank you for pointing out this ablation study in the BitFit paper[1], we missed this particular ablation study when we read the paper. The random row/column update in [1] is indeed similar to our proposal but there are several important differences in terms of motivation and implementation. In terms of motivation the authors of the BitFit paper intend to show for the same number of parameters, updating the bias parameters is better than updating other parameters like random rows/columns. But the main limitation of BitFit is the limited number of bias parameters which makes it difficult to increase the capacity of the finetuning model, and by using row/column updates we can easily increase the capacity of the finetuning model since there are many more row and column parameters than bias parameters. In terms of implementation there are also several differences (with reference to the github BitFit implementation). For example, we just use rows or columns alone without mixing them, we don't update the Layernorm parameters, and we don't use gradient masks in our implementation as it is less efficient.
>
> And as for the difference in performance between BitFit and updating random rows and columns in their ablation studies, we spent some time running the following comparison experiment.
> We run their BitFit github code using the recommended parameters listed in Table 6 of their appendix using bert-base-cased, and we use random seeds 0-4 for the smaller datasets in the GLUE benchmark. Below are the results:
>
> | **Method**           | **CoLA**  | **SST2**  | **MRPC** | **STS-B**  | **RTE**    |
> |----------------------|------------|-----------|-----------|----------|------------|
> | Bitfit |  57.81(1.01)| 90.73(0.24)| 89.92(0.21) | 88.00(0.06) | 70.90(2.59)|
>
> These numbers are a bit lower than those reported in Table 3 of the paper, which could be due to different random seeds used or a different implementation before they release their github code.
>
> Instead of using their random row/column implementation due to efficiency issues with gradient masks, we directly replace the model with our RoCoFT model implementation (also in the submitted supplementary materials) using single row without bias, and we just use their optimization and evaluation code. For hyperparameters we don't do a detailed search but only choose from Table 6 of the BitFit paper or Table 9 of our submission (which was for Roberta). And we obtain the following results:
>
> | **Method**            | **CoLA**  | **SST2**  | **MRPC** | **STS-B**  | **RTE**    |
> |----------------------|-----------|-----------|----------|------------|------------|
> | RoCoFT$_\text{1-row}$   | 56.75(0.59)| 90.78(0.21)| 89.07(0.80) | 86.27(0.22) | 66.20(1.34)|
>
> Apart from RTE and STS-B the results are very similar (within standard error). For RTE we do notice updating bias is better than updating 1 row or 1 column, as it is also seen in Table 1 of our submission (about 1 point difference between BitFit and RoCoFT 1-row or 1-col). For STS-B the difference could be due to difference between bert-base-cased and Roberta. We believe the difference between updating bias and updating 1 row or 1 column is much smaller than the numbers reported in the BitFit paper, either due to the choice of random seeds or the specific implementation on random mixing of rows and columns.
>
> We want to emphasize that we believe BitFit is a very strong method when the number of parameters are limited. In the LoRA paper BitFit is just as good as LoRA of rank 1, and in Table 1 of our submission BitFit has almost the same numbers as RoCoFT 1-row and 1-col, if not better. The main issue is unlike LoRA and our method, it cannot increase its capacity to rank 2, 3, 4 or above to improve its performance.
>
> ## W3. Credit to Bitfit
>  We understand that the  reviewer is concerned with similarities and comparison of the BitFit method with  RoCoFT. However the BitFit  finetuning method freezes all the parameters in weights and classificatoin layers and  finetunes only the additive bias terms. Row/column update is done in table 1 of [1] only as an ablation study and not the main method.  We kindly refer the reviewer to line 92 of manuscript where we cited the BitFit  paper as one of the main related works and Tables 1, 2 and 4, where we compared our method with  the BitFit method.
>
> [1] Zaken, E.B., Ravfogel, S. and Goldberg, Y., 2021. Bitfit: Simple parameter-efficient fine-tuning for transformer-based masked language-models. arXiv preprint arXiv:2106.10199.

---

> > ### Author Response · Authors · 2024-11-23
> > **Response to weakness 4**
> >
> > ## W4. Novelty and Comparison with  random selection
> > We  disagree with the review on performance of row-column updates being as good as random selection. We  kindly refer the  reviewer to tables 1,3,10 and 11 for comprehensive study of this comparison. Please see a vignette below:
> >
> > **BLOOMz$_{7B}$**  ( Tables 3 and  11)
> >
> > | **Method**           | **# TTPs** | **BoolQ** | **PIQA** | **SIQA** | **H.Sw.** | **W.Gra.** | **ARCe** | **ARCc** | **OBQA** | **M.Ar.** | **G.8K** | **A.S.** | **S.eEq** | **S.MP** |
> > |-----------------------|------------|-----------|----------|----------|-----------|------------|----------|----------|----------|-----------|----------|----------|-----------|----------|
> > | RoCoFT$_\text{3-Row}$  | 13.37M     | 66.33     | 74.53    | 73.56    | 56.60     | 72.14      | 73.29    | 57.48    | 72.92    | 79.76     | 70.94    | 70.95    | 70.90     | 54.42    |
> > | 1% of the model parameters selected through uniform sampling | 70.4M      | 65.76     | 74.62    | 73.50    | 56.39     | 72.11      | 72.89    | 56.88    | 72.43    | 79.78     | 71.11    | 70.76    | 70.91     | 54.37    |
> >
> > **Roberta$_\text{Large}$** ( Tables 1 and 10)
> >
> > | **Method**           | **# TTPs** | **CoLA**  | **SST2**  | **MRPC** | **STS-B**    | **QQP**       | **MNLI**      | **QNLI**    | **RTE**    |
> > |----------------------|------------|-----------|-----------|----------|--------------|---------------|---------------|-------------|-----------|
> > | RoCoFT$_\text{3-Row}$ | 0.666M  |67.39 | 96.69| 91.05/92.19| 92.10/92.10 | 90.82/86.11 | 90.98 | 94.85 | 87.83 |
> > | 10% of the model parameters selected through uniform sampling | 35.5M      | 65.32     | 96.59     | 90.93/92.03   | 92.10/92.05   | 90.97/86.78   | 90.89         | 95.06       | 87.91     |
> >
> > In the  tables above, please  note the  significantly lower number of TTPs in  RoCoFT while having competitive or higher accuracies.
> > Moreover, the novelty of  our paper is twofold. First, to the best of our knowledge, the row-column update of the weight matrices despite being starkly low-complexity and   showing competitive accuracies, is not proposed as a method in previous papers. Moreover, explaining the effectivity of fine-tuning method by means of NTK regression is at its infancy in the literature. We believe the NTK regression, which we investigated,  is a powerful tool  for analyzing the learning dynamics of  finetuning.

---

> ### Author Response · Authors · 2024-11-23
> **Response to weakness points 5-6**
>
> ## W5. Finetuning through the lens of NTK regression
> We believe there are several misunderstandings over the purpose of NTK experiments, and we are sorry the brevity of our presentation due to page limits could be responsible for this. We would try to clarify in the following.
>
> 1. It is actually not obvious that the NTK defined by the full parameter set would be similar to the NTK defined by just a few rows/columns, since the NTK of the row/column parameter set is NOT obtained by "changing one row/column" of the NTK of the full parameter set. From the definition of NTK in Section 5 on p7 $K_\theta(x, x') = <\nabla f_\theta(x),  \nabla f_\theta(x)>$,  it is essentially a sum over inner product of gradients. With the NTK over full parameter set it is a sum over gradients of all parameters, and the NTK for row/column is just a sum over gradients of the row/column parameters. The size of the sum is very different. This is reflected by the big difference in magnitude in the NTK values (for example in Figure 4), but when rescaled the similarity patterns over samples defined by the two different NTKs are strikingly similar.
>
> 2. All the NTKs are computed with the pretrained model without any finetuning (noted by line 391 on p8), and we do not compute any NTKs after finetuning. Indeed in Section 5 we do not perform any finetuning at all. The whole point of NTK analysis, as in the original NTK paper, is that neural network training can be approximated by kernel regression using NTK defined by the initial parameters, under the infinite width limit. Malladi et al extends this analysis to the finetuning of LLMs. So if full finetuning can be approximated by kernel regression over the full parameter NTK and RoCoFT can be approximated by kernel regression over the row/column NTK (asymptotically), then the closeness of the full parameter NTK and row/column NTK serves as independent evidence on why full finetuning and RoCoFT can have similar performance. This is our main motivation for exploring NTKs as a way to understand why finetuning with some few parameters work. Also, since NTKs are defined by fixed initial parameters of the pretrained model, good performance of NTK kernel regression (either with the full parameter NTK or row/column NTK) indicates that good features for downstream tasks are already contained in the pretrained model (without further feature learning). This is the other significance of NTK analysis.
>
> In [1,2], it is shown that overparameterized networks  behave linearly around their  initialization, thus
> yielding a model equivalent to learning with positive-definite kernels. In [3],  this phenomenon is further extended to finetuning.
>
> [1] Chizat, L., Oyallon, E. and Bach, F., 2019. On lazy training in differentiable programming. Advances in neural information processing systems, 32.
>
> [2] Jacot, A., Gabriel, F. and Hongler, C., 2018. Neural tangent kernel: Convergence and generalization in neural networks. Advances in  neural information processing systems, 31.
>
> [3] Malladi, S., Wettig, A., Yu, D., Chen, D. and Arora, S., 2023, July. A kernel-based view of language model fine-tuning. In International Conference on Machine Learning (pp. 23610-23641). PMLR.
>
> ## W6. Presentation of the method
>  Section 3 is dedicated to explanation of RoCoFT method. The update weight matrices  $\mathbf{R}$ and $\mathbf{C}$, are introduced in equation (2). Optimal properties of $\mathbf{R}$ and $\mathbf{C}$ such as rank  and robustness to  selection criteria of  rows and columns,  since being numerically  investigated, are  explained in section 6. In particular, we explain in the paper that "Figure 5 presents the comparative results of these four strategies on the SST-2, RTE, QNLI, CoLA, and MNLI datasets for rank $r=4$. Across all datasets, the results show consistent robustness, indicating that our method performs well regardless of the selection criteria—whether based on Max, Min, MinMax, or random selection of rows or columns."

---

> ### Author Response · Authors · 2024-11-23
> **Response to Questions 1-9**
>
> ## Q1.  Notion of 1-row and 1-column
> $RoCoFT_{\textrm{1-Row}}$ and  $RoCoFT_{\textrm{3-Row}}$ respectively  select the first row and first three rows of each  weight matrix for in-place finetuning. In the revised version, we clarified this in line 205 as "RoCoFT$_{r\textrm{-Row(Column)}}$ finetunes the  model according to equation (2), where in $R$ and $C$ the first $r$ rows(columns) are nonzero, respectively".
>
> ## Q2. Typo in RoCoFT 3-row using 5 times the number of parameters as the 1-row
> Thank you pointing this out. There was a Typo in reported TTPs in Table 1, which is corrected in the revised manuscript.
>
> ## Q3.  Comparison with BitFit
> The BitFit Method is included in Tables 1,2 and 4. Please note that Bitfit in terms of number of trainable parameters is only comparable to $RoCoFT_{\textrm{1-Row(Column)}}$. In other words only when  rank is  1. When finetuning with  BitFit, one  cannot scale the number of trainable parameters. Therefore, we mostly used LoRA for comparison.
>
> ## Q4.  Typos
> Thank you for pointing out the typos and grammatical errors. We corrected them.
>
> ## Q5. Abbreviations in the abstract
> In the revised manuscript, we expanded the abbreviations in the abstract.
>
> ## Q6. Selection strategy in the abtract
> In the revised paper we  included this in the  abstract: "Ablation studies are conducted to investigate the impact of different algorithmic choices, including the robustness of RoCoFT to any selection of rows and columns, as well as the optimal rank for the effective implementation of our method".
>
>
> ##  Q7. Notion of a/b for metrics
> Thank  you for  raising this point, we agree that the captions were not clear. In table 1, Accuracy/F1 score for MRPC and QQP is reported and Pearson/Spearman correlations for STS-B.
> In table 2, for SQuADv1.1 and  SQuADv2.0  datasets results are  reported using Exact Match (EM)/F1 scores and  for CNN/Daily Mail datasets results are  reported using ROUGE metrics as  ROUGE-1/ROUGE-2/ROUGE-L. The captions are edited now to clearly reflect the metrics.
>
> ## Q8. Overfitting in  full finetuning
> Yes, we agree that there could be overfitting with FT for some of the smaller datasets like RTE. But we believe this is exactly the reason why we should consider alternatives to full finetuning like many PEFT methods when the downstream task dataset is small. We could apply regularization like early stopping or dropout to FT but this could be more tricky to get right than just directly using PEFT methods.
>
> ## Q9. Optimal rank evaluation not matching Table 1
> Please note that in Table 1 $RoCoFT_{\textrm{1-Row}}$ and $RoCoFT_{\text{1-Column}}$, have 0.083M TTPs comprised of the attention and classifier layer, whereas in Table 7, in order to illustrate the effect of rank, we only   finetune the  attention layers with 0.022M TTPs. Therefore, comparing  the accuracy of SST2  in these two tables is not fair.

---

> ### Comment · Reviewer_cVGh · 2024-11-25
> **Repond to authors**
>
> I thank the authors for their efforts to address my comments and improve my paper. However, I am afraid I still have several concerns.
>
> Typos: The paper still has writing issues, and it would also be much more convenient for the reviewers if the authors had highlighted the changes using some color.
>
> Missing space: benchmarks(Jiang et al., 2024).
> rows(columns)
> Fine-Tuning and Finetuning are used interchangeably
>
> BitFit: It is surprising to me that the authors have not read relevant information from such a closely related work. Although I am not an author of BitFit, I have learned about its details and its relation to this paper since it was cited here. Consequently, I expect the authors to be knowledgeable about this related work.
>
> As I understand from the authors, the main differences between their method and the alternative (or ablation) mentioned in Bit-Fit (which can also naturally scale to more than one row) are three elements: row mixing, layer norm update, and gradient mask. The original version of the paper did not mention any of these details, and even the current version fails to note the latter two as differences from BitFit.
>
> In terms of credit, I do not see adequate acknowledgment given to the ideas already described in Bit-Fit, even if presented as a baseline (or ablation) and not presented as their main method, it was presented in their paper. I highly recommend that the authors revise some sections of the paper to emphasize how they improve upon this idea by identifying the key elements that make it work (including an ablation on all three elements). This is the proper way to convey scientific novelty and contribution.
>
> Why are you choosing the first rows? Doesn't this contradict the entire row selection scheme, which indicates that random is as good as any other selection? I don't understand the issue with what you call row mixing. Is it that rows are mixed between layers?
>
> I would like to thank the authors for their corrections and for clarifying the NTK evaluation. While I find this clarification valuable, my main concerns about the paper remain. I think the underlying idea is interesting, and I believe that improving the presentation of the paper can greatly enhance its quality. I strongly recommend that the authors consider these suggestions to strengthen their work.

---

> ### Author Response · Authors · 2024-11-29
>
> Dear reviewer cVGH
>
> Thank you for the important points you raised.
>
> $\bullet$ We choose the first rows because implementation-wise that’s the simplest. The ablation studies on row or
> column choice just show there are very little effects on finetuning performance among the different choices.
> Therefore we just pick one that’s convenient for implementation.
>
> $\bullet$ As for row mixing, we mean in the ablation
> studies of BitFit, they flip a random coin to decide whether a particular layer use row or column updates.
> So in their rand row/col experiment roughly 50% of their updates are with rows and 50% are with columns,
> when considered across all layers. Our method on the other hand only consider row-only for all layers, or
> column-only for all layers, without mixing their use across layers.
>
> $\bullet$ Some of the additional results and corrections were already added to the appendix and the main body of the paper. Since at this stage it is not possible to upload a revised manuscript with colored editions,  we will
> include the clarifications and replies on additional results in these discussions into the main paper once these
> discussions are finalized.
>
> Sincerely,
>
> Authors

---

> > ### Author Response · Authors · 2024-12-03
> >
> > Dear reviewer cVGh,
> >
> > Thank you for your feedback during the rebuttal and highlighting BitFit as a related work to acknowledge in the paper. BitFit is only one related work and RoCoFT can be considered similar to other PEFT methods already thoroughly discussed in the paper. This is while we did not entirely ignore the BitFit method and it was discussed in  our initial submission. Eventhough  uploading the revised manuscript is not possible at this stage, please note the following revisions that we will make on our paper in the final submission:
> >
> > $\bullet$ Acknowledgement of the BitFit method and stating the limitations of it compared to RoCoFT is definitely our goal in the final  revision of our paper.
> >
> > $\bullet$ The missing spaces in "benchmarks(Jiang et al., 2024)", "rows(columns)", "Fine-Tuning" and "Finetuning" being used interchangeably which require minor revision of the manuscript.
> >
> > We believe the findings of our paper is strong and the PEFT community should know about it.  If there are no remaining technical concerns, we would greatly appreciate your reconsideration of the score. Thank you again for the valuable comments on our work.
> >
> > Sincerely,
> >
> > Authors

---

> > > ### Comment · Reviewer_cVGh · 2024-12-03
> > > **Respond to reviewers**
> > >
> > > I would like to thank the authors once again for their efforts to improve their paper. Unfortunately, as I mentioned earlier, the paper requires substantial revision before it can be considered for publication.
> > >
> > > Specifically, the contribution of this work should focus on differentiating the proposed solution from the one previously presented in BitFit. The current version of the paper does not effectively convey this distinction, and the reader does not gain valuable insights into what the authors did to ensure the success of their method compared to previous approaches.
> > >
> > > As a suggestion, I recommend that the authors emphasize these aspects and consider developing an optimal method for row/column selection that is suitable for this type of fine-tuning.

---

### Official Review · Reviewer_Hy7a · 2024-11-01

**Soundness:** 2
**Presentation:** 3
**Contribution:** 2
**Rating:** 5
**Confidence:** 5

**Summary:**

The paper introduces RoCoFT, a parameter-efficient fine-tuning (PEFT) method designed for large language models (LLMs) that updates only a subset of rows and columns in transformer weight matrices. This approach aims to retain model accuracy while reducing memory and computational requirements compared to traditional fine-tuning methods. RoCoFT achieves state-of-the-art or comparable results on tasks like GLUE, question answering, and summarization, as well as on benchmarks requiring common sense and mathematical reasoning. The authors analyze the method’s effectiveness through neural tangent kernel (NTK) theory, showing that kernels from RoCoFT are numerically close to full-parameter kernels, suggesting that fine-tuning a limited parameter subset preserves core model knowledge.

**Strengths:**

- The presentation is clear, and the paper is easy to follow, with only a few minor typos.
- The proposed method, RoCoFT, is straightforward and demonstrates strong empirical performance.
- The results are reported across multiple tasks and base models, evaluated using various metrics, including memory usage, computation time, and accuracy. This is a good plus to the paper.

**Weaknesses:**

- **Lack of Related Work Discussion**: One weakness of this paper is the limited scope of its related work discussion, focusing primarily on low-rank methods (e.g., LoRA). However, RoCoFT has a closer methodological resemblance to pruning and sparse fine-tuning methods, which are underrepresented in this review. In the parameter-efficient fine-tuning (PEFT) field, methods generally fall into either low-rank or subset of trainable parameter categories, so a more comprehensive comparison should include subset of trainable parameters finetuning baselines (or sparse fine-tuning), such as [1-8]. Adding a discussion of these methods in the related work section would strengthen the contextual foundation of this paper.

- **Need Additional Novelty Clarification**: The paper lacks a detailed discussion of how RoCoFT differs from existing sparse PEFT methods, such as those presented in [1-8].

- **Lack of Baseline Comparisons**: While RoCoFT has similarities with pruning and sparse fine-tuning techniques, the paper currently lacks direct baseline comparisons to these methods. Including baselines from sparse fine-tuning methods in the experiments would offer a more balanced evaluation of RoCoFT's performance and efficiency.

- **Inclusion of More SOTA Models**: The experiments include recent models like DeBERTaV3 and LLaMA-2, which is commendable. However, the study would be more persuasive if it also incorporated newer state-of-the-art models (e.g., Llama3-8B, Llama3.1, Minstrel) to reflect the rapidly advancing field of pre-trained model performance.

- **Typos**:
"prevailing paradiagm" should be corrected to "prevailing paradigm".
"state-of-art" should be "state-of-the-art".
"massive amount of text" should be "massive amounts of text".
"signficant savings" should be "significant savings".

- **Clarity of Baseline Model in Figures**: In Figure 2 and Figure 3 of Section 4, the efficiency comparisons are unclear because the base model for fine-tuning (used to report memory and time costs) is not specified. Similarly, Figure 5 lacks clarity on which base model was used for reporting average accuracy across different metrics. Including these model details would improve transparency in the experimental setup.

[1] The Lottery Ticket Hypothesis: Finding Sparse, Trainable Neural Networks

[2] Parameter-Efficient Fine-Tuning without Introducing New Latency

[3] Sparse Matrix in Large Language Model Fine-tuning

[4] Parameter-Efficient Transfer Learning with Diff Pruning

[5] Training Neural Networks with Fixed Sparse Masks

[6] Scaling Sparse Fine-Tuning to Large Language Models

[7] Composable Sparse Fine-Tuning for Cross-Lingual Transfer

[8] Diff prunning: Parameter-Efficient Transfer Learning with Diff Pruning

**Questions:**

-  **Discussion on Fisher Information**: Reference [5] uses empirical Fisher information to select the most efficient parameters for fine-tuning. It would be beneficial if the authors discussed the efficiency of this method relative to RoCoFT, as this comparison could highlight RoCoFT’s strengths and potential trade-offs.

-  **Memory Cost Clarification**: In fig.2, the author reports the memory cost for baselines and RoCoFT. However, the results are not easy to follow/understand. In LLM, since Adam optimizer is the most common optimizer, the memory cost for full Adam optimizer will be 2 times than the model weights. For instance In Llama-2-7B model, the model weight is 13.6G and the optimizer will cost 2*13.6GB. However, LoRA can reduce the optimizer memory cost to less than 1%. In Fig.2, the authors report the memory cost for RoCoFT and LoRA is still 2 times than the model weight, can you kindly discuss why is that?

- **percentage of trainable parameters**: In PEFT field, papers usually use percentage of trainable parameters to present the algorithm efficiency. Since in Figure2, Figure3 of Section4, efficiency comparison, the author didn’t clarify what is the fine-tuning base model author used to report all the memory and time cost. It’s also different to find In figure what is the fine-tuning base model author used to report the average accuracy for different metrics in figure 5. Can the author discuss the percentage of trainable parameters they use?

- **Implementation for Memory Reduction**: Low-rank methods like LoRA use additional trainable adapters, while sparse fine-tuning often applies binary masks to reduce memory. It would strengthen the paper if the authors elaborated on how the paper implement RoCoFT to achieve memory reduction and speedup compared to these existing techniques, and discuss it from the aspect of system. Do RoCoFT need to implement full forward and backward propagation for all parameters? Do RoCoFT will introduce more modules during the fine-tuning process?

I would like to discuss the questions I raised regarding the weaknesses and concerns with the authors. If my concerns are adequately addressed, I would be willing to reconsider my rating.

**References**:

[1] The Lottery Ticket Hypothesis: Finding Sparse, Trainable Neural Networks

[2] Parameter-Efficient Fine-Tuning without Introducing New Latency

[3] Sparse Matrix in Large Language Model Fine-tuning

[4] Parameter-Efficient Transfer Learning with Diff Pruning

[5] Training Neural Networks with Fixed Sparse Masks

[6] Scaling Sparse Fine-Tuning to Large Language Models

[7] Composable Sparse Fine-Tuning for Cross-Lingual Transfer

[8] Diff Pruning: Parameter-Efficient Transfer Learning with Diff Pruning

---

> ### Author Response · Authors · 2024-11-22
> **Response to weakness points 1-3**
>
> We thank  the reviewer for their detailed feedback recommendations for our work. We will be happy to discuss any addtional  questions the reviewer may have.
>
> ## W1. Related Work Discussion
> Thank you for highlighting the need for a broader discussion of related work. We agree that while RoCoFT shares similarities with low-rank methods like LoRA, it also has similarities with pruning and sparse fine-tuning techniques
> We will expand the related work section to include more relevant papers from the sparse fine-tuning and pruning categories, as suggested. We appreciate your guidance and references, which will help in strengthening the contextual foundation of the paper.
>
> ## W2. Novelty Clarification
> We thank the reviewer for the suggested list of references. We have indeed overlooked the discussion on sparse fine-tuning methods in our literature review in related works. Below is our view on how our method relates to the sparse fine-tuning methods listed in [1-8], which we will also include in the updated related works section in our paper. Thank you so much for helping us improve this aspect of our paper.
>
> "Apart from low-rank adaptor methods Sparse Fine-Tuning is another group of PEFT methods that focuses on directly training only a very small subset of model parameters during finetuning. Sparsity can be achieved in two different ways, either by pruning after full finetuning or by selecting a sparse set of masks to train before finetuning. Diff pruning[4/8] encourages sparsity by using L0-norm regularization during finetuning, while [7] makes use of the Lottery Ticket Hypothesis[1] to prune the weights after full finetuning. Unlike our proposed method they both require computational costs close to full finetuning. [3] selects submatrix blocks as masks using maximal gradient change during warmup as criterion, while [5] selects masks based on Fisher information. Both require some precomputation before a sparse mask can be selected for finetuning. [2] selects unimportant weights for task-agnostic finetuning while [6] propose a finetuning method that can adaptively grow or shrink the sparsity pattern during finetuning. Unlike our method which uses only rows and columns, these sparsity masks can be unstructured patterns and less efficient in actual implementation. Our method can be seen as belonging to both low-rank adaptor methods and sparse fine-tuning, as with few rows or columns chosen the updates are naturally both low-rank and sparse."
>
> ## W3. Baseline Comparisons
> We thank you for the valuable feedback. We recognize the importance of comparing RoCoFT directly with sparse fine-tuning baselines to present a more balanced evaluation of its performance and efficiency, though we have tried to compare it with some recent state-of-the-art methods.
>
> In response to your suggestion, we have included a new comparison with recent works, including sparse fine-tuning methods. Additionally, we will incorporate further baseline comparisons with well-established sparse fine-tuning methods in our experiments to strengthen the evaluation.
>
> | Dataset | LoRA-XS[9] | Vera[10] | LoRAFA[11] | SFT[12] | Diff Pruning[8] | FSM[1] | RoCoFT (row) | RoCoFT (column) |
> |---------|------------|---------|-----------|--------|-----------------|--------|--------------|-----------------|
> | SST2    | 93.19      | 93.89   | 93.65     | 94.28  | 93.77           | 94.11  | 94.92        | 94.69           |
> | CoLA    | 58.49      | 60.35   | 60.49     | 64.45  | 62.45           | 62.77  | 63.53        | 62.95           |
> | MNLI    | 85.34      | 85.64   | 86.11     | 86.64  | 85.32           | 85.85  | 86.73        | 86.76           |
> | QNLI    | 90.42      | 90.22   | 91.42     | 92.11  | 92.14           | 91.81  | 92.12        | 91.89           |
>
>
> [1] The Lottery Ticket Hypothesis: Finding Sparse, Trainable Neural Networks
>
> [2] Parameter-Efficient Fine-Tuning without Introducing New Latency
>
> [3] Sparse Matrix in Large Language Model Fine-tuning
>
> [4] Parameter-Efficient Transfer Learning with Diff Pruning
>
> [5] Training Neural Networks with Fixed Sparse Masks
>
> [6] Scaling Sparse Fine-Tuning to Large Language Models
>
> [7] Composable Sparse Fine-Tuning for Cross-Lingual Transfer
>
> [8] Diff Pruning: Parameter-Efficient Transfer Learning with Diff Pruning
>
> [9] LoRA-XS: Low-Rank Adaptation with Extremely Small Number of Parameters.
>
> [10] Vera: Vector-based random matrix adaptation.
>
> [11] Lora-fa: Memory-efficient low-rank adaptation for large language models fine-tuning.
>
> [12] Scaling Sparse Fine-Tuning to Large Language Models

---

> ### Author Response · Authors · 2024-11-22
> **Response to weakness points 4-6**
>
> ## W4. More SoTA Models
> Thank you for pointing this out. We appreciate the suggestion to include newer state-of-the-art models, such as Llama3-8B, Llama3.1, to better align with the rapid advancements in pre-trained models. While our experiments currently focus on widely used models like DeBERTaV3 and LLaMA-2 to establish the effectiveness of RoCoFT, we acknowledge that incorporating newer models would further strengthen the study by demonstrating its scalability and applicability to the latest architectures. Since the suggested newer models were only a few months old when we submitted our paper and there were not a lot of published results on them from the PEFT literature compared to LLaMA-2 or DeBERTaV3, we decided to go with the older models given our computational resources constraints.
> In future work, we plan to extend our evaluations to include these emerging models. However, it is important to note that resource constraints, such as computational requirements and availability of pretrained checkpoints, can impact the feasibility of incorporating newer models within the current scope. Nonetheless, we are committed to adapting RoCoFT to the most recent developments in the field to ensure it remains relevant and competitive.
>
> ## W5. Typos
> Thank you for pointing out the typos. We corrected them.
>
> ## W6. Clarity of Baseline Model in Figures
> Thank you for this helpful observation. We used the RoBERTa-base model as the base model for fine-tuning in Figures 2, 3, 4, and 5, with a batch size of 32. We will update the figure captions and descriptions in Section 4 to clearly indicate the base model and batch size used for reporting memory, time costs, and accuracy metrics, improving the transparency of our experimental setup.

---

> > ### Author Response · Authors · 2024-11-22
> > **Response to questions**
> >
> > ## Q1. Fisher Information
> > Reference [5] uses Fisher information to select the most important features for finetuning, and other works like [7,3] try to select the most important features to finetune via weight magnitudes or largest gradient change.
> > On the other hand, reference [2] takes a different view and selects the unimportant(unused) features for finetuning to different tasks. Interestingly both approaches work on the typical benchmark datasets considered. This is corroborated by our ablation studies on the choice of rows or columns to finetune, which we showed relatively little difference in choosing the most important or most unimportant rows/columns to finetune when scored by the pruning criterion used in WANDA. Therefore we don't make any effort in trying to select the "best" rows/columns to finetune in RoCoFT and just take the first few rows/columns in the weight matrices. This is the main difference with methods that selects the best features to finetune using criterions like Fisher information. It is possible for extreme cases of sparsity (e.g. much fewer than the ~1M trainable parameters in Table 1) that these sparse fine-tuning methods based on feature/masks selection can outperform the feature/mask-agnostic approach used in RoCoFT, but for typical benchmark datasets we do see little difference.
> >
> > ## Q2. Memory Cost Clarification
> > Thank you for this insightful feedback. In Figure 2, we used the same experimental setup as in Table 1, where RoCoFT is set to rank 1, while other LoRA-based methods are at rank 2, resulting in LoRA’s rank being effectively 20 times that of RoCoFT. For a fairer comparison, we then adjusted all LoRA-type methods to rank 10 and found that the memory costs were approximately 3.79 GB for LoRA and 3.57 GB for RoCoFT.
> >
> > We updated Figure 2 to reflect this fair comparison, showing memory costs after one epoch of full training using SST-2. This will clarify the memory cost differences across methods under comparable conditions.
> >
> > ## Q3. Percentage of trainable parameters
> > Thank you for your feedback. In our paper, we reported the total trainable parameters rather than a percentage. The reason is that RoCoFT does not introduce additional adapter parameters, whereas other methods require adapters that increase the total number of parameters, thus altering the percentage of trainable parameters. Since adapter size varies across methods, using total trainable parameters allows for a more direct and consistent comparison of algorithm efficiency.
> >
> > To address the base model clarity in Figures 2, 3, and 5, we used RoBERTa-base for fine-tuning and will update the figure captions accordingly to ensure transparency in our experimental setup.
> >
> > ## Q4. Implementation for Memory Reduction
> >  Thank you for your questions regarding memory reduction and speedup in RoCoFT. Our implementation achieves memory efficiency without introducing additional modules, as we only update a subset of parameters within the existing model structure. Specifically, RoCoFT replaces the layers in the pretrained model with custom modules that update selected rows or columns, managed with nn.Linear(). We split the weights into trainable and non-trainable portions. Only a subset (rank k) of the original weight matrix is marked as trainable, while the remaining parameters are moved to a buffer (non-trainable). This avoids memory overhead from additional trainable adapters or binary masks and enables parameter-specific updates without dynamically creating new tensors. The non-trainable weights are detached and stored in buffers to save memory, ensuring no gradients are computed or stored for these weights. This approach allows us to avoid the full memory cost typically incurred with trainable adapters like in LoRA. During training, RoCoFT does not require full forward and backward propagation for all parameters. Instead, the concatenation of trainable and non-trainable weights is performed once and stored, so the model does not introduce additional computational modules or overhead during the fine-tuning process.

---

> ### Author Response · Authors · 2024-11-30
>
> Dear reviewer Hy7a,
>
> We are grateful for your constructive feedback, which has greatly contributed to improving the quality of our work.
> We would be happy to address any remaining concerns you may have regarding the revised manuscript and the new experimental results. Please feel free to provide further comments or suggestions, and we will make every effort to incorporate them promptly and thoroughly. We look forward to your feedback.
>
> Sincerely,
>
> Authors

---

> ### Author Response · Authors · 2024-12-02
>
> Dear reviewer Hy7a:
>
> Thank you for valuable  recommendations about presentation of our work. Please note that incorporating all the requested changes from reviewers could exceed the page limit, especially for related works that we cannot put into the appendix. We don’t want our paper to be rejected during the review stage due to page limit issues. Therefore we chose to include the answers in this discussion thread and extra tables and results in the appendix before the discussions are finalized. Sorry for not having included these changes directly into the paper last week but unfortunately we cannot modify the pdf at this stage. We are grateful for your suggested changes especially the list of extra references which is very helpful. We will include them in revisions of our paper.
>
> **W1-2:** We will add  this to the introduction:
>
> "Apart from low-rank adaptor methods Sparse Fine-Tuning is another group of PEFT methods that focuses on directly training only a very small subset of model parameters during finetuning. Sparsity can be achieved in two different ways, either by pruning after full finetuning or by selecting a sparse set of masks to train before finetuning. Diff pruning[4/8] encourages sparsity by using L0-norm regularization during finetuning, while [7] makes use of the Lottery Ticket Hypothesis[1] to prune the weights after full finetuning. Unlike our proposed method they both require computational costs close to full finetuning. [3] selects submatrix blocks as masks using maximal gradient change during warmup as criterion, while [5] selects masks based on Fisher information. Both require some precomputation before a sparse mask can be selected for finetuning. [2] selects unimportant weights for task-agnostic finetuning while [6] propose a finetuning method that can adaptively grow or shrink the sparsity pattern during finetuning. Unlike our method which uses only rows and columns, these sparsity masks can be unstructured patterns and less efficient in actual implementation. Our method can be seen as belonging to both low-rank adaptor methods and sparse fine-tuning, as with few rows or columns chosen the updates are naturally both low-rank and sparse."
>
> **W4:** While, we agree applying RoCoFT on newer pretrained models  e.g., Llama3-8B, Llama3.1, Minstrel should be conducted in  future works, the current results on Roberta, DeBERTaV3, BART, BLOOMz-7B, GPT-J-6B, LLaMA-2-7B, LLaMA-2-13B aligns well with the benchmarks commonly used in the PEFT literature. We believe these results are strong and the PEFT community  should know about it.
>
> **W6, Q3:**  This requires minor  edition on  captions of figures 2,3,4 and 5 to  reflect the trained model and fine-tuning  configurations.
>
> **Q4:** We will add this in the appendix:
>
> RoCoFT replaces the layers in the pretrained model with custom modules that update selected rows or columns, managed with nn.Linear(). We split the weights into trainable and non-trainable portions. Only a subset (rank k) of the original weight matrix is marked as trainable, while the remaining parameters are moved to a buffer (non-trainable). This avoids memory overhead from additional trainable adapters or binary masks and enables parameter-specific updates without dynamically creating new tensors. The non-trainable weights are detached and stored in buffers to save memory, ensuring no gradients are computed or stored for these weights. This approach allows us to avoid the full memory cost typically incurred with trainable adapters like in LoRA. PyTorch pseudocode for replacing a linear layer in transformer is as below
> ```python
> class RoCoFTRow(nn.Module):
>     # inputs: F is the Linear Layer to be converted
>     #         r is the rank (number of rows/columns to be selected)
>     #         use_bias decides whether to train bias term
>     def __init__(self, F, r, use_bias):
>         # Set rows 1 to rank r as trainable weights
>         self.trainable_W = nn.Parameter(F.weight[:r, :].clone())
>
>         # Set rows r and above as non-trainable and move to buffer
>         self.register_buffer('non_trainable_W', F.weight[r:, :].clone().detach())
>
>         # Handle bias
>         if F.bias is not None:
>             self.bias = nn.Parameter(F.bias.clone().detach(), requires_grad=use_bias)
>         else:
>             self.bias = None
>
>     def forward(self, x):
>         full_weight = torch.cat([self.trainable_W, self.non_trainable_W], dim=1)
>         out = torch.nn.functional.linear(x, full_weight, self.bias)
>         return out
> ```
> The version for columns are implemented similarly.  During training, RoCoFT does not require full forward and backward propagation for all parameters. Instead, the concatenation of trainable and non-trainable weights is performed once and stored, so the model does not introduce additional computational modules or overhead during the fine-tuning process.

---

### Note · Authors · 2025-01-26

I have read and agree with the venue's withdrawal policy on behalf of myself and my co-authors.